# Photodynamic Effectiveness of Copper-Iminopyridine Photosensitizers Coupled to Zinc Oxide Nanoparticles Against *Klebsiella pneumoniae* and the Bacterial Response to Oxidative Stress

**DOI:** 10.3390/ijms26094178

**Published:** 2025-04-28

**Authors:** Dafne Berenice Hormazábal, Ángeles Beatriz Reyes, Matías Fabián Cuevas, Angélica R. Bravo, David Moreno-da Costa, Iván A. González, Daniel Navas, Iván Brito, Paulina Dreyse, Alan R. Cabrera, Christian Erick Palavecino

**Affiliations:** 1Laboratorio de Microbiología Celular y Fotodinámica, Centro de Investigación en Ingeniería de Materiales, Facultad de Medicina y Ciencias de la Salud, Universidad Central de Chile, Lord Cochrane 418, Santiago 8330546, Chile; dafne.hormazabal@alumnos.ucentral.cl (D.B.H.); angeles.reyes@alumnos.ucentral.cl (Á.B.R.); maticuep@uc.cl (M.F.C.); angelica.bravo@ucentral.cl (A.R.B.); 2Departamento de Química Inorgánica, Facultad de Química y de Farmacia, Pontificia Universidad Católica de Chile, Vicuña Mackenna 4860, Macul, Santiago 7820436, Chile; drmoreno@uc.cl; 3Departamento de Química, Facultad de Ciencias Naturales, Matemática y del Medio Ambiente, Universidad Tecnológica Metropolitana, Las Palmeras 3360, Ñuñoa, Santiago 7800003, Chile; igonzalezp@utem.cl; 4Departamento de Química, Facultad de Ciencia, Universidad de Chile, Las Palmeras 3425, Ñuñoa, Santiago 7800003, Chile; daniel.navas@edu.udla.cl (D.N.); paulinadreyse@uchile.cl (P.D.); 5Departamento de Química, Facultad de Ciencias Básicas, Universidad de Antofagasta, Av. Angamos 601, Antofagasta 1270300, Chile; ivan.brito@uantof.cl

**Keywords:** photodynamic therapy, multi-drug resistance, copper(I) complex, ZnO nanoparticles, *Klebsiella pneumoniae*

## Abstract

One of the most urgent threats to public health worldwide is the ongoing rise of multidrug-resistant (MDR) bacterial strains. Among the most critical pathogens are MDR-*Klebsiella pneumoniae* strains. The lack of new antibiotics has led to an increased need for non-antibiotic antimicrobial therapies. Photodynamic therapy (PDT) has become increasingly significant in treating MDR bacteria. PDT uses photosensitizer compounds (PS) that generate reactive oxygen species (ROS) when activated by light. These ROS produce localized oxidative stress, damaging the bacterial envelope. A downside of PDT is the limited bioavailability of PSs in vivo, which can be enhanced by conjugating them with carriers like nanoparticles (NPs). Zinc nanoparticles possess antibacterial properties, decreasing the adherence and viability of microorganisms on surfaces. The additive or synergistic effect of the combined NP-PS could improve phototherapeutic action. Therefore, this study evaluated the effectiveness of the copper(I)-based PS CuC1 compound in combination with Zinc Oxide NP, ZnONP, to inhibit the growth of both MDR and sensitive *K. pneumoniae* strains. The reduction in bacterial viability after exposure to a PS/NP mixture activated by 61.2 J/cm^2^ of blue light photodynamic treatment was assessed. The optimal PS/NP ratio was determined at 2 µg/mL of CuC1 combined with 64 µg/mL of ZnONP as the minimum effective concentration (MEC). The bacterial gene response aligned with a mechanism of photooxidative stress induced by the treatment, which damages the bacterial cell envelope. Additionally, we found that the PS/NP mixture is not harmful to mammalian cells, such as Hep-G2 and HEK-293. In conclusion, the CuC1/ZnONP combination could effectively aid in enhancing the antimicrobial treatment of infections caused by MDR bacteria.

## 1. Introduction

In the 21st century, the threat of returning to a pre-antibiotic era is one of the most urgent public health problems worldwide due to the continued increase in multidrug-resistant (MDR) bacterial strains [1,2]. As bacteria have progressively accumulated more and more resistance factors, we have fewer antimicrobial options for effective treatment. This is why the World Health Organization (WHO) and the US Centers for Disease Control and Prevention described the situation as a global crisis and an impending catastrophe of returning humans to the pre-antibiotics era [3]. Due to the limited development of new antimicrobial drugs in 2017, the WHO updated a global priority list of antibiotic-resistant bacteria. Bacteria on this list are a priority in developing new antimicrobial options [4,5]. *Klebsiella pneumoniae*, extended-spectrum β-lactamase (ESBL) producers, and carbapenemase (KPC) are among the most relevant bacterial strains in the WHO ranking [4,5]. *K. pneumoniae* can be found asymptomatically in healthy carrier individuals’ intestinal tract, skin, nose, and pharynx [1,6]. However, its opportunistic pathogenic behavior positions it as one of the most frequent causative agents of infections in the elderly, adults, and children [7]. *K. pneumoniae* is the agent of pneumonia and urinary tract infections that can be mild or complicated, resulting in severe disease with mortality between 30 and 70% [8,9]. Also, MDR strains of *K. pneumoniae* are one of the most prevalent healthcare-associated infections (HAIs) [10], where the only antibiotics therapeutic options to treat KPC strains are limited to polymyxins (such as colistin) and tigecycline [11]. Unfortunately, the prevalence of polymyxin-resistant bacteria has significantly increased since 2016 due to the indiscriminate use of colistin in farms [3,8].

The antimicrobial crisis can be solved through complementary therapies to antibiotic therapy [12]. For example, antimicrobial photodynamic therapy (PDT) has gained significant relevance for treating infections with MDR bacterial strains [13]. The PDT is based on photosensitizer compounds (PS) that produce local oxidative stress when activated by light [14]. Photosensitizers capture the quantum energy of light and transfer it to oxygen present in an aqueous solution, where reactive oxygen species (ROS) are generated. Energy transfer can occur in two ways: accompanied by an electron, called Type I, or without an electron, called Type II [14]. Type I ROS are superoxide (O_2_^•−^), hydrogen peroxide (H_2_O_2_), and hydroxyl radicals (^•^OH). Although the Type II effect only produces singlet oxygen (^1^O_2_), the effect may last longer as the electrons are not transferred [14]. Oxidative stress may damage the bacterial envelope, increasing its permeability and producing a non-specific lethal effect on bacteria such as *K. pneumoniae* strains [15]. One of the significant advantages of antimicrobial photodynamic therapy is its broad spectrum of application and non-specificity, which minimizes the development of resistance [16]. However, the lack of specificity of classical organic photosensitizers results in several limitations, such as photobleaching, the need for proximity to the target microorganism, low singlet oxygen quantum yield, and reduced dispersion in water.

Among the most used inorganic PS are the second- and third-row transition metal complexes, such as Ru, Rh, Ir, or Pt. These emerged to overcome limitations due to their improved photophysical properties and promising photodynamic activities [17,18]. However, the disadvantages of using complexes of these metals include their potential toxicity to normal cells, lack of biocompatibility, and high cost. Due to these disadvantages, the research on complexes from the first row, such as Ti, Mn, Fe, Cu, or Zn, has gained significant attention recently [19]. Overall, the studies indicate that copper complexes are promising photosensitizers with favorable photophysical properties, such as strong light absorption in the visible regions, high singlet oxygen generation, and suitable excited state lifetimes [18]. Besides, the photophysical properties of copper compounds are easily tuned by ligand design [20,21]. Even so, the main drawback of copper complexes is their stability in biological media.

On the other hand, the use of nanoparticles as carriers has been studied to address the limitations shown by molecular complexes [22]. Nanoparticles are a type of material whose dimensions vary between 1 and 100 nm. The biocidal efficacy of nanoparticles is due to their small size and the large contact surface, which allows for a more intimate interaction with the microbial structures. For example, Zinc oxide nanoparticles (ZnONP) exhibit antibacterial activity and can reduce the adherence and viability of microorganisms on biomedical surfaces [23]. Combining the photosensitizers with nanoparticles could significantly improve their antimicrobial photodynamic activity; hence, the therapeutic action can be enhanced by combining additive or synergistic effects of both materials [22]. Combining 2 μg/mL of the photosensitizers with 64 μg/mL of ZnO nanoparticles may improve the production of various ROS types, including the Type I (O_2_^•−^) or the Type II (^1^O_2_) effects. The generation of these ROS will depend on the surface area per unit mass of ZnONP; the more significant the surface area is, the greater the production of ROS [23]. The impact of PDT using photosensitizers and nanoparticles (NPs) has been investigated in vitro on both Gram-positive and Gram-negative bacteria [24] (Table 1). Non-biodegradable NPs, such as gold, offer biocompatibility and adjustable sizes for drug delivery [25,26]. Due to their lack of biological degradation, photosensitizers are released along with the diffusion of singlet oxygen upon light activation [24]. Gold nanoparticles (AuNPs) conjugated with methylene blue (MB-AuNPs) achieved a 96% eradication rate of S. aureus [27]. Zirconium (Zr)-based metal-organic NPs, modified with titanium (Ti) to boost ROS generation, resulted in nearly 100% inhibition of *S. aureus* [28]. Dextran-capped MB-AuNPs linked to concanavalin A (Con A) targeted *K. pneumoniae*, achieving 97% eradication through lectin-mediated adhesion to bacterial cell walls following light activation [29].

As a Gram-negative bacterium, *K. pneumoniae* presents several common virulence factors with Enterobacteria, such as fimbriae, capsule, biofilm, and survival to oxidative stress, among others [1]. Regarding survival from oxidative stress, each microorganism has developed complex antioxidant defense systems that combine enzymatic and non-enzymatic mechanisms to prevent the effects of ROS exposure. These mechanisms are gene-encoded factors that can be expressed under defined conditions. For example, in Gram-negative bacteria, the response to the stress of Type I ROS is well characterized; the *oxyR* gene is related to resistance to H_2_O_2_ [12,30], and *sodABC* is related to resistance to O_2_^•−^ and OH^−^ [31]. On the other hand, there are insufficient biochemical studies characterizing the response of Gram-negative bacteria to ROS produced by the Type II effect (^1^O_2_), the most common photooxidative stress [32]. The response to Type II ROS is best characterized in the photosynthetic bacterium *Rhodobacter sphaeroides* and points to the regulon controlled by the alternative sigma factor RpoE [32,33] and the posttranscriptional factor hfQ [34]. The *rpoE* and *hfq* genes are activated when damage occurs in the cell envelope and may control the expression of genes such as *mrkD*, *magA*, *acrA*, *rpoE* and *hfQ* [12,32]. All those genes have been associated with the bacterial response to stress caused by photodynamic therapy [35,36,37]. This work investigated the antimicrobial capacity of copper(I) based photosensitizer compounds coupled to Zinc nanoparticles. These copper(I) compounds correspond to [(*N*,*N*)Cu(*P*,*P*)]^+^ heteroleptic complexes with *N*,*N* ligands derived from 4-cyanophenyl-iminopyridines and DPEphos as the *P*,*P* auxiliary ligand. We evaluated the ability of these systems to reduce bacterial viability after light activation, expressed in colony-forming units (CFU) per mL. Additionally, the minimal effective concentration and the cytotoxicity for mammalian cells caused by the PDT were determined. We also evaluated the mode of action of PDT by determining how it modifies the gene expression of factors associated with the response to oxidative stress and the recovery of cell envelope structures.

## 2. Results

### 2.1. Synthesis and Structural Characterization of Photosensitizer and Nanoparticles Mix

#### 2.1.1. Synthesis and Structural Characterization of Photosensitizers

As mentioned in the introduction, copper complexes are attractive photosensitizer compounds [18,20,21]. Accordingly, in this work, these were designed based on our previous reports on photoactive Cu(I) complexes [38,39,40]. The synthesis of [(*N*,*N*)Cu(DPEphos)]PF_6_ (CuC1–3) is shown in Figure 1A, where *N*,*N* correspond to 4-cyanophenyl-iminopyridines derivatives and DPEphos to the auxiliary diphosphine ligand. The formation of the complexes corresponds to a ligand substitution reaction between the acetonitrile molecules in the metal precursor and the *N*,*N* and *P*,*P* ligands. The formed complexes differ on the alkyl substituents at the 2,6 positions of the aromatic ring (shown as R in Figure 1A). The CuC1 compound has H as the substituent, CuC2 has methyl and CuC3 has isopropyl. The structures of the CuC1–3 was established based on their spectral properties (NMR, FT-IR, and EA; see Experimental Section and *ESI*). X-ray diffraction was also performed on CuC3, and the molecular structure is shown in Figure 1B.

#### 2.1.2. Photophysical Characterization of Photosensitizers and Nanoparticles Mix

The CuC1–3 complexes were characterized by UV-Vis spectrophotometry in acetone as the solvent (cutoff at 330 nm) at 2 mg/mL concentration. The resulting UV-Vis spectra are presented in Figure 1C. All complexes exhibit absorption bands below 350 nm, attributed to spin-allowed ligand-centered (^1^LC) transitions involving the *N*,*N* and *P*,*P* ligands. Additionally, a tail extending to longer wavelengths can be ascribed to spin-allowed metal-to-ligand charge transfer (^1^MLCT) transitions [39,40]. As shown in Table 2, the photosensitizer compounds show absorption bands close to 421 and 460 nm. The photosensitizer incorporating ZnONP displays similar spectral behavior to CuC1, following the protocol described in Section 2.6. The UV-Vis spectrum of ZnONP in ethanol shows a prominent absorption band near 280 nm, with no significant absorption observed beyond 300 nm (Figure 1C). Additionally, Table 1 shows the recorded calculated quantum yield (F_em_) for PS control (PSRu-L3, 0.156) and for CuC1 (0.038). The PDT activity of the CuC1 compound is comparable with the antimicrobial activity reported for PSRu-L3 [35].

#### 2.1.3. Synthesis of ZnONP

The synthetic procedure carried out comprises the addition of K_2_CO_3_ to Zn^+2^ to produce hydrozincite (Zn_5_(CO_3_)_2_(OH)_6_) that corresponds to a 3:2 stoichiometric ratio of Zn(OH)_2_ and ZnCO_3_. With the addition of acetic acid, Zn(CH_3_COO)_2_ is produced, and then with the hydrolysis of Zn(CH_3_COO)_2_ in ethanol/water, adding NaOH was possible to form ZnO. Then, the solvothermal treatments yielded ZnONPs, an effective procedure to suppress and control the crystal growth and facilitate ZnO dispersion [41]. The ZnO nature of the solid obtained was corroborated by XRD diffraction (Figure 1D) compared with the XRD pattern ZnO hexagonal wurtzite lattice [42], identifying similar signals. TEM images of ZnO structures obtained made it possible to identify the particles at the nanometric scale range of 10–15 nm, in a pseudo-ellipsoidal shape, distributed in organized conglomerates well compactly (Figure 1E). Some exfoliated nanoparticles appear unlike what has been identified in ZnO nanoparticle clusters with looser structures [43]. In addition, the ZnO NPs clusters are proposed to be generated from small nanoparticles smaller than 10 nm, which does not apply in this case. A large surface-area–volume ratio characterizes these NPs and may act as a favorable carrier in suspension in aqueous solutions, where they are expected to work [44,45].

### 2.2. In Vitro Bacterial Susceptibility to Antibiotics

The bacterial susceptibility was assayed in vitro by the Kirby–Bauer methodology using a diffusion disc in a Petri dish for the ESBL characterization of the KPPR-1 and ST258 *K. pneumoniae* strains. The strains grown in LB broth were suspended in ca-MH broth at an OD_600_ of 0.08–0.13 nm, equivalent to 0.5 MacFarland. Lawn seeding on MH Agar was used to place the antibiotic discs and incubated at 37 °C for 16–20 h, following CLSI guidelines [46]. Figure 2 confirms that the KPPR-1 strain is susceptible to ceftazidime (Caz), Caz + clavulanic acid (Cla), Ceftriaxone (Cro) and Cro + Cla by showing an inhibition halo > 5 mm (Figure 2A). On the other hand, it is confirmed that strain ST258 is not susceptible to Caz or Cro since it does not present an inhibition halo but shows sensitivity to Caz + Cla and Cro + Cla when presenting a halo > 5 mm (Figure 2B). The susceptibility profile of strain ST258 confirmed it phenotypically as ESBL according to the standard CLSI protocol. The non-ESBL and ESBL status were subsequently confirmed genotypically by the multiplex PCR assay, which identified the associated carrier genes [47]. Figure 2C shows that strain KPPR-1 harbors only the SHV gene, whereas strain ST258 harbors both the SHV and TEM genes.

### 2.3. Antimicrobial Photodynamic Activity of Copper-Based Compounds

We first analyzed the PDT activity of three photosensitizer (PS) compounds based on copper oxide formed by the ligands L1 (CuC1), L2 (CuC2) and L3 (CuC3). The purified powder of the PS compounds was dissolved in acetone at a concentration of 2 mg/mL and prepared for PDT and PS control groups in an aqueous solution at 16 µg/mL. The PDT activity was tested in vitro*,* assessing bacterial viability by inhibiting the growth of the *K. pneumoniae* KPPR-1 and ST258 strains. Bacteria were used at a concentration of 1 × 10^7^ UFC/mL, and all groups were mixed in triplicate with the PSs in PBS 1x. The PDT group was immediately exposed to 10 min of blue light fluency at 61.2 J/cm^2^, while the control PS groups were kept in the dark. After incubation, bacteria were recovered, and viability was determined through serial micro-dilutions and plating for colony counts. The Atlanta CDC and CLSI advise that antimicrobials must reduce bacterial load by more than 99.9%, equivalent to 3 log_10_ [46], to claim a practical antibacterial effect. Figure 3A,B demonstrate that the CuC1 PS compound significantly reduced bacterial viability from 5.6 × 10^7^ to 7.3 × 10^4^ CFU/mL (~3 log_10_) for the KPPR-1 strain and from 4.3 × 10^7^ to 6.3 × 10^4^ CFU/mL (~3 log_10_) for the ST258 strain (*p* > 0.05). The other compounds, CuC2 and CuC3, showed no significant reduction in bacterial viability after photodynamic activation. The control ligand (Lig) also exhibited no photodynamic antimicrobial activity.

Then, we searched for the minimal effective concentration (MEC) of the CuCl compound to inhibit bacterial growth by at least 3 log_10_. The MEC for the CuCl PS was determined by incubating 1 × 10^7^ CFU/mL of the KPPR-1 or ST258 *K. pneumoniae* strains solubilized in ca-MH, with PS concentrations ranging from 0 to 32 µg/mL, activated for 10 min with 61.2 J/cm^2^ of blue light fluency. After treatment, viable bacteria were counted using a serial micro-dilution method and colony counts on ca-MH agar plates. As shown in Figure 3C, compared to the untreated control (0 PS), the MEC for CuCl for both bacterial strains were determined to be 4 µg/mL (*p* < 0.01), where bacterial viability decreased below 3 log_10_. Higher photosensitizer concentrations did not significantly enhance effectiveness, as demonstrated by the comparison of bacterial viability between 4 and 32 µg/mL (*p* > 0.05).

### 2.4. Photosensitizer Coupled to Zn Oxide Nanoparticles

The CuC1 activity reduces bacterial viability by 3 log_10_, the minimum required to classify it as a photosensitizer [46]. Activity that, although not optimal for treatment, does not increase significantly with an increase in PS concentration. Other authors have improved the photodynamic activity of PS by mixing them with several compounds, such as nanoparticles [48]. We first mixed 1 × 10^7^ CFU/mL of *K. pneumoniae* strains with concentrations of ZnONP in the 0 to 128 μg/mL range in PBS 1x and subjected it for 10 min to 61.2 J/cm^2^ fluency. As shown in Figure 4A, although a significant decrease in bacterial viability occurred, even at the highest concentrations of the nanoparticles, it did not reach 3 log_10_ reduction. Then, we chose to start with 80 μg/mL nanoparticles to be mixed with the CuC1 PS following the research done by the group of Garin et al., 2021 [22]. An initial mixture of 4 µg/mL of the CuC1 compound and 80 µg/mL of zinc oxide nanoparticles were combined with 1 × 10^7^ CFU/mL of *K. pneumoniae* strains KPPR-1 and ST258. As indicated above, a PDT group was subjected for 10 min to 61.2 J/cm^2^ fluency, and a PS group was kept in the dark for the same period. Figure 4B shows that the ZnONP significantly increased (*p* < 0.001) the bactericidal capacity of the CuC1 photosensitizer, reducing the viability of the KPPR-1 and the ST258 strains >5 log_10_ compared to the untreated controls. We then determined at what concentration the zinc nanoparticle shows bactericidal photodynamic activity.

### 2.5. Determination of the Minimum Inhibitory Concentration

It has evaluated the best proportion of photosensitizer and nanoparticles that maintain the reduction of >5 log_10_. The minimum effective concentration of the photosensitizer compound CuC1 in a fixed concentration of 80 µg/mL ZnONP was determined to inhibit the growth of *K. pneumoniae* KPPR-1 and ST258 strains. PDT was performed with serial dilutions of 0–8 µg/mL of CuC1 and activated with 61.2 J/cm^2^ of light fluence. As seen in Figure 5A, the minimum effective concentration of the CuC1 was reduced from 4 to 2 µg/mL when mixed with 80 µg/mL of ZnONP. We then determined if the nanoparticle could be used at lower concentrations, keeping the CuC1 MEC. Then, a fixed amount of 2 µg/mL of the CuC1 was mixed with serial dilutions of 0–128 µg/mL of ZnONP and the bacteria *K. pneumoniae* KPPR-1 and ST258 strains were subjected to PDT. As shown in Figure 5B, at 64 µg/mL, the ZnONP retains its ability to increase the bactericidal photodynamic activity >5 log_10_ of 2 µg/mL CuC1. Higher concentrations of ZnONP, such as 80 or 128 µg/mL, maintains baseline bacterial growth inhibition. Then, the MEC for the photosensitizer/nanoparticle mixture is 2 μg/mL CuC1 with 64 μg/mL ZnONP.

### 2.6. Determination of the Adsorption Efficiency

Once the optimal PS/NP ratio was identified (1:32), the adsorption of the CuC1 compound onto the ZnO nanoparticles was determined by mixing 2 µg/mL CuC1 with 64 µg/mL ZnONP (PS/NP mix). The PS/NP mix was incubated at 25 °C for 10 and 20 min, stirring every minute, and then centrifuged at 14,000 rpm for 1 min before measurement. The measurement was conducted using a UV-1900 Shimadzu (Kyoto, Japan) at an absorbance of 380 nm and recorded before and after incubation to determine the adsorption efficiency. The CuC1 absorbance at 380 nm was 2711 before incubation, 2382 after 10 min, and 2365 after 20 min. Using the formula AE = (C0 − C1)/C0 × 100, the adsorption efficiency after 10 and 20 min was 12.1% and 12.8%, respectively. This adsorption demonstrated that PS CuC1 interacts with the NPs, improving their ability to inhibit bacterial growth by more than 5 log_10_ (Figure 4). This improvement relies on photodynamic activation, as the PS/NP mix is inactive in the absence of light.

### 2.7. Evaluation of the Mode of Action of PDT Using the PS/NP Mix

To explore the mode of action of the 2 µg/mL CuCl + 64 µg/mL ZnONP mix (PS/NP mix), we attempted to identify whether the produced ROS were Type I or Type II. Then, we determined how much H_2_O_2_ and ^1^O_2_ were produced because of PDT. Specific solution probes for H_2_O_2_ or ^1^O_2_ were employed to differentiate between the various ROS. These probes do not react with other ROS, such as hydroxyl radical (^•^OH) or superoxide (O_2_^•−^) [49]. Furthermore, we evaluated the response of the treated bacteria, which is also specific for Type I or Type II ROS [50].

We used the ROS-Glo H_2_O_2_ assay (Promega Corporation, Madison, WI, USA) to measure the H_2_O_2_ produced by the PS/NP mix and compared it to a standard curve constructed with serial dilutions (0.05 to 1.6 μM) of hydrogen peroxide (Sigma-Aldrich, St Louis, MO, USA). The PS/NP mix was added with 10 μL of H_2_O_2_ substrate and exposed to PDT for 10 min at 61.2 J/cm^2^. The serial dilution of oxygen peroxide was also mixed, and all samples were incubated for 60 min at room temperature. After incubation, ROS-Glo^TM^ detection solution was added to each well, and after a 20 min incubation, the relative luminescence values (RLU) were measured. The singlet oxygen sensor green (SOSG) probe was used to measure the ^1^O_2_ produced by the PS/NP mix and compared it to a standard curve constructed with serial dilutions (2.5 to 20 μM) of methylene blue (MB). The MB and the PS/NP mix in PBS solution were mixed with 1 μM SOSG reagent and exposed to blue light for 10 min at 61.2 J/cm^2^. Fluorescence produced by the generation of ^1^O_2_ was measured with excitation/emission at 488/525 nm.

As shown in Figure 6A, the serial dilutions create a standard curve that display luminescence dependent on hydrogen peroxide concentration, with a mean maximum of 1200 RLU at 1.6 µM hydrogen peroxide. Figure 6B illustrates that the light-activated PS/NP mix (PDT) yielded a mean luminescence of 8200 RLU, comparable to 0.025 µM hydrogen peroxide. The unactivated control (PS) produced readings below the detection limit of the technique, similar to the ROS-Glo background. As indicated in Figure 6C, the MB produces a concentration-dependent fluorescence standard curve with a mean maximum of 4300 a.u. at 20 µM MB. Figure 6D reveals that, compared to the unactivated control (PS), the light-activated PS/NP mix (PDT) generated a high fluorescence level comparable to that of 20 µM MB, suggesting significant production of ^1^O_2_. This production of ^1^O_2_ is prompted by the PS/NP mix, as the SOSG yields only background fluorescence. Thus, the combination of low H_2_O_2_ production with high ^1^O_2_ generation indicates that the CuCl/ZnONP photodynamic system primarily induces Type II ROS, establishing it as the main mechanism for ROS generation.

Our previous results show that metal-based photosensitizer compounds such as Ir(III) [51], Re(II) [36], and Ru(II) [35] induce a transcriptional response compatible with Type II ROS production. Thus, we determined whether treatment with our Cooper(I)-based photosensitizer CuC1 enhanced by ZnO nanoparticles induces a similar transcriptional response pattern. We evaluated the bacterial-specific transcriptional response of treated bacteria compared to untreated bacteria, as we did before [50]. Then, total RNA was obtained from bacteria growing in the log phase (OD_6000_ 0.4–0.6 nm), separated into untreated control group and PDT treated group with 2 µg/mL CuC1 mixed with 64 µg/mL ZnONP, PS/NP mix).

In *E. coli*, the response to oxidative stress caused by Type I ROS, such as H_2_O_2_, is controlled by the OxyR transcription factor [51] and by superoxide for the *sodA*BC operon that encodes the multimeric enzyme superoxide dismutase (SOD). Figure 7A shows that compared to the untreated control, the PDT with the PS/NP mixture induced a significant (*p* < 0.05) downregulation of the *oxyR* gene in both the KPPR-1 (0.142-fold) and the ST258 (0.075-fold) strains of *K. pneumoniae*. The transcriptional response of the *sodA* gene was mixed when comparing both strains; the PDT induced the downregulation (mean 0.6-fold, *p* > 0.05) in the KPPR-1 strain and an overexpression (1.7-fold, *p* < 0.05) in the ST258 strain (Figure 7A). In the *R. sphaeroides* bacteria, the extra-cytoplasmatic factor *rpoE*, as well as the *hfq* posttranscriptional regulator genes, are overexpressed upon ^1^O_2_ stressor. RpoE controls several genes that specifically respond to ^1^O_2_ stress and are posttranscriptional modulated by the Hfq factor. Like *R. sphaeroides*, the transcriptional response to photooxidative stress caused by Type II ROS seems to be driven by the RpoE regulon and the action of the posttranscriptional regulator HFQ [32]. In Figure 7A, the results indicate that, compared to control bacteria, photodynamic treatment with the PS/NP mixture induced significant (*p* < 0.01) overexpression of the *rpoE* gene in KPPR-1 (46-fold) and ST258 (45-fold) strains. Consistently, PDT also induced significant (*p* > 0.05) overexpression of the *hfq* gene in both bacterial strains (7.5- and 7.4-fold, respectively). Consequently, the lack of regulation of the *oxyR* and *sodA* genes, coupled with the upregulation of *rpoE* and *hfq*, corresponds with the bacterial response to Type II photooxidative stress. This confirms that singlet oxygen production is the primary reactive oxygen species (ROS) generated by CuCl/ZnONP photodynamic therapy.

Our results suggest that the type II mechanism predominates over the type I mechanism, characterized by a strong overexpression of the regulatory factor RpoE. In *K. pneumoniae,* the RpoE transcriptional factor controls the expression of extracytoplasmic factors such as the MrkD, MagA, RmpA and AcrB proteins [12,33]. As shown in Figure 7B, in the KPPR-1 and ST258 strains, PDT significantly (*p* < 0.05) overexpressed (mean of 8- to 15-fold) the bacterial cell envelope maintenance *mrkD*, *magA* and *rmpA* genes. However, the efflux pump gene *acrB* showed mixed results between the two strains studied, with no significant overexpression (mean 1.1 ± 2SD-fold) in the KPPR-1 strain and a significant (*p* < 0.05) downregulation (0.64-fold) in the ST258 strain. The overexpression of these factors suggests that damage occurs in the cell envelope structures that the bacteria attempt to replace.

### 2.8. Determination of Cytotoxicity on Mammalian Cells of the CuC1 + ZnONP Mix

The compounds are designed for the treatment of active *K. pneumoniae* infections. Therefore, the CuC1 + ZnONP mixture was tested in vitro to confirm its compatibility with tissues, ensuring that it does not cause damage to eukaryotic cells upon contact. The human hepatocellular Hep-G2 and the human kidney HEK2-93 cell lines were used to test in vitro two parameters of cytotoxicity: cell death and cell survival. Cells were grown in a 10 cm culture plate with DMEM + 10% FBS and 5% CO_2_ atmosphere. When the cell culture reached a 70–90% confluence, the cell culture medium was removed, and the cells were washed once with 1x D-PBS. For the treatment, the semiconfluent cultures were incubated with 2 µg/mL CuC1 + 64 µg/mL ZnONP mix (PS/NP mix) in 1x D-PBS, a group was kept in the dark (PS/NP) and a group was treated with 61.2 J/cm^2^ of blue light (PDT). A control group (Crtl) with no PS/NP mix was included.

To evaluate cell death, the DMEM + 10% FBS medium was replenished after treatment and the cells were incubated for an additional 24 h at 37 °C with 5% CO_2_. As shown in Figure 8A, no signs of cytotoxicity were observed in the treated Hep-G2 or HEK293 cells compared to control cells. Additionally, no dark cytotoxicity was noted in cells treated with the PS/NP mix but not exposed to light. To quantify cytotoxicity, the cells were trypsinized, combined with trypan blue and the dead cells were counted in a hemocytometer chamber. As shown in Figure 8B, when Hep-G2 or HEK293 cell lines were exposed to the PS/NP mix in the dark or activated by light, the number of live cells was not significantly reduced (ns = *p* > 0.05, Student’s *t*-test comparing treated cells with the controls of the untreated cells). A colony-forming assay was conducted to determine cell viability [52]. Following treatment, cells were trypsinized and an aliquot containing 1000 Hep-G2 or 500 KEK293 cells was seeded into a 6-well plate containing DMEM + 10% FBS medium and incubated for five to seven days at 37 °C with 5% CO_2_. Once colonies formed, the medium was removed and the cells were fixed with methanol and stained with crystal violet. Colonies consisting of 50 or more cells were recorded, and cell viability was calculated as colonies formed over seeded cells, comparing treated with untreated cells. As shown in Table 3, the plating efficiency (PE) was determined to be 17% and 22.3% for Hep-G2 and HEK293 of the control cells, respectively. Table 3 and Figure 8C present cell viability expressed as the surviving fraction (SF: mean ± SD) of treated cells. There was no significant reduction in the number of viable cells for liver Hep-G2 or kidney HEK2-93 cell lines exposed to the PS/NP mixture, whether in the dark or activated by light.

## 3. Discussion

*Klebsiella pneumoniae* has long been recognized as a pathogen (Carl Friedländer first described it as a cause of pneumonia in 1882) and remains one of the most common nosocomial pathogens in the world [9]. Although *Klebsiella pneumoniae* is mainly found in healthy carriers’ gastrointestinal tract and the nasopharynx, it is a significant source of pneumonia and urinary tract infections [53]. The World Health Organization recognizes ESBL- and KPC-producing strains of *K. pneumoniae* as critical threats to public health [2,54]. The constant increase in the incidence of MDR *K. pneumoniae* strains generates a significant decrease in therapeutic alternatives [4]. Complementary therapies such as PDT may be helpful as they can act as rescue therapies, reverse antibiotic resistance and reduce the spread of MDR bacteria [55]. PDT may complement conventional antimicrobial therapies to treat infections with MDR *K. pneumoniae* strains [12]. We previously demonstrated in vitro photosensitizer’s advantage in inhibiting the bacterial growth of MDR *K. pneumoniae* strains [35,36,56].

The limitations of classical organic photosensitizer complexes drive us to explore other candidates, such as our first-row transition metal complexes based on copper(I), which show enhanced photophysical properties [17,18]. From a structural viewpoint, compounds CuC1–3 possess an *N*,*N* ligand that is easily synthesized and has an elevated steric hindrance tunability around the metal center. Additionally, the cyano group in the phenyl ring is an anchor group that interacts with different Lewis acid sites, facilitating an interaction with nanoparticle surfaces [57]. The DPEphos ligand was selected due to the metal center’s elevated stability and the complex’s improved photophysical properties [38]. The straightforward and economical synthetic path allowed us to obtain the copper(I) complexes in good yields (85–96%). The NMR characterization of these compounds agrees with a mononuclear metal center surrounded by the *N*,*N* ligand and the diphosphine (see Experimental Section and *ESI*). In addition, the single crystal XRD of CuC3 agrees with the NMR characterization, crystallizing in the monoclinic space groups *P*2_1_/*n* and showing a distorted tetrahedral geometry around the copper(I) center, with a τ4′ value of 0.77 [58]. The antimicrobial effect observed for the PS/NP mixture can, at first glance, be attributed to the characteristics of the photosensitizer’s excited states. The excited state of CuC1 exhibits a significant contribution from ^3^MLCT, demonstrating an emission quantum yield comparable to that of PSRu-L3. This similar quantum yield may account for the comparable antimicrobial photodynamic activity observed in the PS/NP mixture and PSRu-L3 [35].

From the three coopers (I)-based PSs tested, only the CuC1 shows an acceptable antimicrobial PDT activity, decreasing the bacterial viability >3 log_10_ (Figure 3). Although this activity is the minimum acceptable, it may not help treat active infections in vivo. This activity may be influenced by steric hindrance around the metal center. The CuC1 complex experiences the least steric hindrance, featuring a hydrogen substituent, while CuC3 displays the greatest steric effect due to the presence of isopropyl groups on the aromatic ring. The ability of the complex to undergo geometric changes during Cu(I)/Cu(II) interconversion is determined by the structural rigidity around the metal center [59]. Complexes with lower structural rigidity experience less resistance to geometric change, leading to lower redox potentials, which directly influence the single-electron transfer [60]. Furthermore, increasing the steric bulk of the ligand around the metal center may impact the efficiency of single-electron transfer [61]. Then, we decided to test if including nanoparticles in the mixture improves the antimicrobial activity of PS. Also, working as a carrier, the nanoparticles may help to overcome other PS limitations, such as photobleaching or the need for proximity. The addition of nanoparticles to the PS complexes has previously been used to improve the pharmacokinetic properties of PSs [62]. The nanoparticles may interact physically with the PS to stabilize vibrational states, enhancing the intrinsic photodynamic properties [63] or acting as carriers that improve the PS delivery [64]. We confirmed through the adsorption efficiency determination that 12% of the photosensitizer molecules physically interact with the nanoparticles. The test was conducted at the PS/NP ratio with the highest antimicrobial effectiveness. We also observed that extending the incubation time beyond 10 min at room temperature showed no further improvement in adsorption. Room temperature was chosen for the test, aligning with the conditions used for PDT experiments and reflecting how the mixture would be prepared for clinical application.

ZnO nanoparticles have been used synergistically with PSs, which efficiently induce ROS in cancer cells [43]. For example, Zinc nanoparticles have been functionalized with 5-aminolevulinic acid (ALA), working as carriers delivering within tumor cells [64]. These Zn nanoparticles disassemble, releasing large amounts of zinc ions (Zn^2+^) that promote the conversion of the ALA to protoporphyrin IX (PpIX) [64]. The PpIX is the photosensitizer active form of the ALA, which also shows antimicrobial photodynamic properties [65]. Other authors have used cationic derivatives of Zn(II) phthalocyanine as PS, which showed acceptable antimicrobial activity against *E. coli* and *K. pneumoniae* [66]. Also, the ZnONP is a secure material as it is shown to have no toxicity in many applications in cosmetics, medicine, deodorants and sunscreens, among others [67]. Therefore, we tested zinc oxide-based nanoparticles to improve the photodynamic properties of the CuC1 compound. The ZnONP was shown to interact with the CuC1 PS, increasing its capacity to inhibit bacterial growth >5 log_10_ (Figure 4). This improvement depends on photodynamic activation, as the PS/NP mix is inactive without light. The need for light activation can be sustained when treating superficial infections such as surgical wounds (*K. pneumoniae* is one of the main causes) or other skin infections. For pneumonia or urinary tract infections (UTI), the PS/NP mix can be activated on-site using devices such as optical fibers delivered through a catheter, which transmit light into internal hollow organs, including non-spherical cavities [68,69,70]. While this therapy cannot treat bacteremia or bloodstream infections, it can significantly assist in eliminating the source of the infection, such as a UTI. Through serial dilutions, we determined the minimal effective concentration and the optimum ratio of PS and NP at 2 µg/mL CuC1 and 64 µg/mL ZnONP (1:32 ratio) (Figure 5). These quantities are in the range of other antimicrobial compounds, such as antibiotics of clinical use to treat active infections [46]. Experimental studies [71] suggest that PS-loaded ZnO nanoparticles show higher PDT activity than free PS, as demonstrated in this work (Figure 5). The interaction between CuC1 and ZnONPs leads to enhanced stability and antimicrobial activity through multiple physicochemical and biological mechanisms. In terms of stability, ZnONPs function as a protective scaffold that prevents CuC1 from aggregating or degrading. This occurs through electrostatic interactions, hydrogen bonding, or coordination between CuC1 and the hydroxyl-rich surface of ZnONPs [72]. By altering the electronic environment of CuC1, ZnONPs can stabilize their oxidation states (Cu(I)/Cu(II)), which is crucial for maintaining their redox activity. Beyond stability, the synergistic antimicrobial activity of the CuC1-ZnONP system is primarily driven by enhanced reactive oxygen species (ROS) generation and bacterial membrane disruption. Their combination may further improve photo-Fenton-like reactions, producing more efficient and sustained ROS [73]. Together, these factors render CuC1-ZnONP composites highly effective and durable antimicrobial agents [74].

To explore the mode of action of this PS/NP mixture, we determined the reactive oxygen species generated (Figure 6) and the type of transcriptional response produced by the bacteria (Figure 7). The detection of H_2_O_2_ and ^1^O_2_ was achieved with the ROS-Glo assay and the SOSG probes, respectively. Both assays are specific for the mentioned molecules inducing ROS, without cross-reactivity with other active molecules, such as hydroxyl radical (^•^OH) or superoxide (O_2_^•−^) [49]. The fit of the PS/NP mixture readings with the calibration curve to produce H_2_O_2_ yielded a value of 0.0125 μM, presenting no significant differences with the background measurements. On the other hand, to determine the production of ^1^O_2_, a calibration curve with the well-known photosensitizer methylene blue (MB) was generated. The fitting for the obtained PS/PN mixture values showed significant singlet oxygen generation, comparable with 20 μM of MB. This reading was the maximum concentration at which we observed fluorescence linearity (Figure 6A). These results strongly suggest that the induced ROS effect predominantly follows a Type II mechanism due to the meager production of H_2_O_2_ and the significant generation of ^1^O_2_.

Afterward, we used the bacterial transcriptional response to corroborate the produced ROS type. The low response of the *oxyR* and *dosA* genes suggests insignificant production of peroxides and hydroxyl anions to which the bacteria was exposed [30,31]. Thus, the levels of H_2_O_2_ induced by PDT may not be high enough to fully induce the transcriptional response to Type I ROS, meaning the participation of Type II ROS. Since oxidative stress induced by Type II ROS is not well characterized in *K. pneumoniae*, the photooxidative stress model described in *Rhodobacter sphaeroides* [32] was used to differentiate the transcriptional response (Figure 7A). In consequence, the upregulation of the *rpoE* and *hfq* genes aligns with the bacterial response to type II photooxidative stress. We have seen similar behavior before when bacteria are treated with photodynamic compounds where the bacterial transcriptional response reveals greater exposure to singlet oxygens than to peroxides [35,36].

Damage to the bacterial envelope by the photooxidative stress derived from the generation of ROS may be responsible for inhibiting bacterial growth [15]. Type II ROS response genes, such as *rpoE* and *hfq*, upregulated in PDT-treated bacteria, coordinate the envelope stress response. Genes of the RpoE operon and the small RNAs modulating *hfq* genes respond in *Rhodobacter sphaeroides* to the membrane photooxidative stress caused by singlet oxygen [32]. The observed overexpression of membrane factors such as the *mrkD*, *magA* and *rmpA* genes (Figure 7B), suggests that in *K. pneumoniae*, they are under the control of the RpoE operon and probably regulated by small RNAs that modulate the *hfq* gene in a similar way to what occurs in *R. sphaeroides* [32,75,76]. However, we could not verify that the *acrB* gene is also involved. Our data strongly suggest that the bacteria are responding to photooxidative aggression that deteriorates structures of its envelope, such as membrane, cell wall, or capsule. Therefore, damage to the cell envelope must be the primary mechanism causing bacterial death triggered by PDT. Furthermore, PDT acts through the type II oxidative stress mechanism, from which bacteria cannot recover.

Finally, we needed to evaluate the safety of the compound by analyzing evidence of cytotoxicity in mammalian cells because PDT induces nonspecific photooxidative stress, which can disrupt the normal metabolism of eukaryotic host cells. As previously observed, mammalian cells tolerate the exposure to ^1^O_2_ well, led by the PS/ZnoNP mix, as we observe very low or absent compound-induced cytotoxicity. We used hepatocyte-derived Hep-G2 cell lines from human sources to assess any potential liver toxicity, a crucial organ for drug detoxification [77]. We also did not see cytotoxicity in human kidney-derived HEK293 cells, confirming the absence of a potential renal effect that could explain impaired drug elimination [78]. Given that the antimicrobial activity of PDT occurs immediately after light exposure, we replicated the same conditions for the eukaryotic cells. Images of cultured cells show no evidence of cytopathic damage, nor does there appear to be any loss of cell viability in trypan blue exclusion assays or colony formation assays.

Both Type I reactive oxygen species (ROS), like the hydroxyl radical (OH^⋅^), and Type II ROS, such as singlet oxygen (^1^O_2_), are highly reactive entities that can cause significant damage to cells. However, the OH^⋅^ is generally considered more toxic to both mammalian cells and bacteria [79]. For mammalian cells, OH^⋅^ is one of the most reactive ROS; even in small amounts, it may cause extensive damage to lipids, proteins and DNA, resulting in immediate apoptosis or necrosis. Although ^1^O_2_ is toxic for both, mammalian cells have more robust antioxidant systems (e.g., superoxide dismutase, catalase and glutathione) that help them mitigate the damage [80]; then, it can be managed by eukaryotic cellular antioxidant defenses [81]. Because the ^1^O_2_ tends to react more with lipids and proteins, it can be highly toxic for bacteria, causing oxidative damage to the bacterial envelope components such as the cell membrane, capsule proteins and also DNA [82]. Therefore, the bactericidal effect of photooxidative stress, through the generation of ^1^O_2_, does not affect mammalian cells.

## 4. Materials and Methods

### 4.1. Synthesis and Photophysical Characterization of the Photosensitizer Compounds

#### 4.1.1. General Information and Materials

All commercially available reagents and solvents were used as received unless otherwise specified. The purity of the reagents is as follows: 4-(6-Formylpyridin-2-yl) benzonitrile (97%), aniline (99%), 2,6-dimethylaniline (98%), 2,6-diisopropylaniline (97%), [Cu(CH_3_CN)_4_]BF_4_ (97%) and DPEphos (98%). The ligands L1–3 were synthesized according to the reported procedures [83]. All complexes were characterized by spectroscopic and spectrometric analyses. NMR spectra were recorded on NMR Bruker AV 400 MHz. The chemical shifts were expressed as parts per million relatives to TMS (^1^H and ^13^C, δ(SiMe_4_) = 0) or an external standard [δ(CFCl_3_) = 0 for ^19^F NMR and δ(H_3_PO_4_) = 0 for ^31^P NMR]. Additional 2D experiments supported most NMR assignments. FT-IR spectra were recorded on a Shimadzu Tracer 100 spectrophotometer using KBr pellets. The elemental analysis was performed using Elementary Analysis System GmbH, Model Vario EL III. The organic compounds were purified using a chromatography column (Merck silica gel 60 (70–230 mesh)). NIR spectra were registered using an Agilent Technologies UV-Vis-NIR spectrophotometer. UV-Vis-spectra was registered using a Shimadzu UV-1900 spectrophotometer. Photoluminescence spectra were taken on an RF-600 Shimadzu Spectrofluorophotometer. Time-resolved luminescence data were taken on a laser flash photolysis apparatus comprised of a Continuum Surelite II Nd: YAG laser (excitation at 355 nm, FWHM = 6–8 ns, was provided by THG from the 1064 nm fundamental). An entrance slit of a 300 mm focal length Acton SpectraPro 2300i emitted by the sample triple grating, flat field and double exit monochromator equipped with a photomultiplier detector (Hamamatsu R3896, Hamamatsu Photonics, Hamamatsu, Japan). The photomultiplier’s signals were processed using a Teledyne LeCroy 604Zi (400 MHz, 20 GS/s) digital oscilloscope. The emission quantum yields were obtained using the PSRu-L3 (Φ_em_ = 0.156 in acetonitrile solution) compound [35] as the standard by the comparative method [84]. Crystallographic data were collected at 293(2) K on a Bruker D8 Venture diffractometer equipped with a Photon-III C14 detector, using graphite monochromated CuKα (λ = 1.54178 Å) radiation. The diffraction frames were integrated using the APEX4 package (Bruker, S.; SAINT, S. Bruker AXS Inc., Madison, WI, USA, 2002) and were corrected for absorption with SADABS, Univ-Göttingen (Sheldrick, G.M. correction of area detector data Software, Sadabs 1996). The intrinsic phasing of SHELXT [85] solved the crystal structures of the title compounds using the OLEX2 software [86] and refined them with full-matrix least-squares methods based on F^2^ (SHELXL) [87]. Full details can be found in the Electronic Supporting Information (ESI) and the independently deposited crystallography information files (CIF), CCDC number 2,377,087 for CuC3 (Appendix A).

#### 4.1.2. Synthesis and Characterization of the Copper(I) Compounds CuC1–3

In a glass vial, one equiv. of [Cu(CH_3_CN)_4_]BF_4_ (0.36 mmol), one equivalent of DPEphos (0.36 mmol) and one equivalent of the corresponding ligand L1–3 was added (0.36 mmol). Then, the vial was sealed with a septum and was nitrogen flushed for 5 min. Later, 3 mL of anhydrous dichloromethane was added through a purged syringe. The reaction mixture was stirred at room temperature for 1 h under a nitrogen atmosphere. Afterward, the volatiles were removed under reduced pressure and the crude product was purified by crystallization from a mixture of CH_2_Cl_2_ and toluene at −20 °C.

Complex CuC1. Orange-colored solid, 96% yield. ^1^H NMR (400 MHz, CDCl_3_, 298 K): δ/ppm 8.69 (s, 1H), 8.23 (d, *J* = 4.4 Hz, 2H), 7.67 (t, *J* = 4.6 Hz, 1H), 7.36 (dt, *J* = 16.5, 7.5 Hz, 4H), 7.25–7.16 (m, 10H), 7.09 (t, *J* = 7.4 Hz, 1H), 7.02 (d, *J* = 8.1 Hz, 2H), 6.97–6.89 (m, *J* = 2.4 Hz, 10H), 6.76 (broad, 4H), 6.69 (d, *J* = 7.9 Hz, 2H), 6.57 (m, 2H), 6.49 (d, *J* = 7.8 Hz, 2H). ^13^C(^1^H) NMR (101 MHz, CDCl_3_, 298 K): δ/ppm 162.4, 158.8, 156.8 (t, *J*^C-P^ = 5.7 Hz), 151.1, 148.8, 143.2, 139.9, 133.8, 133.2 (dt, *J*^C-P^ = 11.7, 7.7 Hz), 132.2, 130.7 (d, *J*^C-P^ = 7.1 Hz), 130.6 (m), 129.8 (t, *J*^C-P^ = 16.6 Hz), 129.2, 128.8, 128.6, 127.4, 124.8, 122.9 (t, *J*^C-P^ = 15 Hz), 121.4, 119.6, 118.1, 112.8. ^19^F NMR (400 MHz, CDCl_3_, 298 K): δ/ppm −73.00 (d, *J*^F-P^ = 712 Hz). ^31^P(^1^H) NMR (160 MHz, CDCl_3_, 298 K): δ/ppm −13.82, −144.21 (hept, *J*^P-F^ = 712 Hz). FT-IR (KBr): n/cm^–1^ 2230 (C≡N), 1589 (C=N), 1222 (O-C), 841 (PF_6_). Elemental analysis (C_55_H_41_CuN_3_OP_2_): calc: C 64.11; H 4.01; N 4.08; O 1.55. Found: C 65.94; H 4.15; N 4.02; O 1.62.

Complex CuC2. Orange-colored solid, 85% yield. ^1^H NMR (400 MHz, CDCl_3_, 298 K): δ/ppm 8.36 (m, 1H), 8.34 (s, 1H), 8.01 (d, *J* = 7.6 Hz, 1H), 7.93 (d, *J* = 7.9 Hz, 1H), 7.43–7.31 (m, 6H), 7.25 (broad, 1H), 7.19–6.95 (m, 3H), 6.85 (broad, 1H), 6.76 (broad, 10H), 6.64 (d, *J* = 7.9 Hz), 6.44 (broad, 4H), 1.56 (s, 6H). ^13^C(^1^H) NMR (101 MHz, CDCl_3_, 298 K): δ/ppm 166.4, 159.0, 156.5 (broad), 150.6, 148.7, 142.6, 140.3, 134.1 (d, *J*^C-P^ = 6.2 Hz), 133.6 (broad), 132.5, 132.3 (broad), 130.5 (m), 130.2, 129.7 (m), 129.6, 129.2 (broad), 129.0, 128.6 (broad), 128.5, 128.1, 126.2, 122.8 (t, *J*^C-P^ = 14.7 Hz), 119.4, 119.1, 111.4. ^19^F NMR (400 MHz, CDCl_3_, 298 K): δ/ppm −73.69 (d, *J*^F-P^ = 713 Hz). ^31^P(^1^H) NMR (160 MHz, CDCl_3_, 298 K): δ/ppm −14.58, −144.29 (hept, *J*^P-F^ = 713 Hz). FT-IR (KBr): n/cm^–1^ 2228 (C≡N), 1589 (C=N), 1220 (O-C), 841 (PF_6_). Elemental analysis (C_57_H_45_CuF_6_N_3_OP_3_): calc: C 64.68; H 4.29; N 3.97; O 1.51. Found: C 62.62; H 4.49; N 3.83; O 1.58.

Complex CuC3. Orange-colored solid, 93% yield. ^1^H NMR (400 MHz, CDCl_3_, 298 K): δ/ppm 8.48 (t, *J* = 7.8 Hz), 8.33 (s, 1H), 8.03 (d, *J* = 8.0 Hz, 1H), 7.94 (d, *J* = 7.6 Hz, 1H), 7.50–7.43 (m, 2H), 7.35 (broad, 9H), 7.17 (broad, 4H), 7.09–6.99 (m, 4H), 6.89–6.83 (m, 10H), 6.71–6.62 (m, 2H), 6.41–6.36 (m, 4H), 2.74–2.64 (m, 2H), 0.63 (dd, *J* = 28.5, 6.7 Hz, 12H). ^13^C(^1^H) NMR (101 MHz, CDCl_3_, 298 K): δ/ppm 166.3, 159.0, 156.6 (broad), 150.4, 147.2, 141.6, 140.8, 139.0, 134.15 (t, *J*^C-P^ = 8.3 Hz), 133.8 (broad), 132.7, 131.9, 130.7 (broad), 130.6 (broad), 130.4, 129.7, 129.5 (broad), 128.3, 127.4, 124.9, 124.0, 123.2 (broad), 119.7 (broad), 118.7, 112.3, 28.3, 25.8, 21.5. ^19^F NMR (400 MHz, CDCl_3_, 298 K): δ/ppm −73.22 (d, *J*^F-P^ = 713 Hz). ^31^P(^1^H) NMR (160 MHz, CDCl_3_, 298 K): δ/ppm −14.78, −144.25 (hept, *J*^P-F^ = 713 Hz). FT-IR (KBr): n/cm^–1^ 2223 (C≡N), 1589 (C=N), 1221 (O-C), 841 (PF_6_). Elemental analysis (C_61_H_53_CuF_6_N_3_OP_3_): calc: C 65.74; H 4.79; N 3.77; O 1.44. Found: C 68.52; H 4.93; N 3.64; O 1.41.

### 4.2. Synthesis and Characterization of the Nanoparticles

The synthesis of the ZnO NPs was developed according to the literature description [41]. The Zn^2+^ precursor solution (25 mL of ZnSO_4_⋅7H_2_O 0.050 M) was stirred and heated to 80 °C, and then 25 mL of 0.15 M aqueous K_2_CO_3_ solution was added under constant stirring. A white precipitate was observed after four hours of stirring at 80 °C. It was filtered, washed several times with water and dried at 80 °C. Then, 25 mL of 7.0 M aqueous acetic acid solution was added, stirring the mixture for 15 min. The solvent was removed by evaporation, and the solids were dissolved in 50 mL of an ethanol/water (1:1) mixture and precipitated using 25 mL of 0.10 M NaOH. The white solid precipitate was centrifuged, washed with deionized water several times and dried at 220 °C for six hours in an oven. Then, the nanoparticles of ZnO were prepared from the solid obtained in a solvothermal process, where the solid was suspended in an ethylene-glycol: water (10:1 mL) mixture in a round bottom flask and kept under constant stirring at 80 °C; then, the suspension was transferred to a stainless-steel reactor and treated with solvothermal synthesis for six hours at 200 °C. The obtained nanoparticles were cleaned several times with water and acetone and finally dried at 80 °C. The structural characterization of ZnONP was performed using a SIEMENS D 5000 X-ray diffractometer (Cu Kα, λ ¼ 1.5418 Å, operation voltage 40 kV, current 30 mA). The morphological characterization was made using a high-resolution transmission electron microscope Hitachi HT7700 (Hitachi, Tokyo, Japan). The sample was previously suspended in absolute ethanol, dropped on a TEM grid and left to dry at room temperature.

### 4.3. Determination of Reactive Oxygen Species Production of PS/NP Mix

The ROS-Glo^TM^ H_2_O_2_ Assay kit (Promega Corporation, Madison, WI, USA) was used to measure the H_2_O_2_ produced by PDT with the PS/NP mix in combination with 10 μL of H_2_O_2_ substrate. A standard curve was constructed with H_2_O_2_ (MERCK) from 0.05–1.6 μM in PBS. The singlet oxygen sensor green (SOSG) reagent (Lumiprobe, Hunt Valley, MD, USA) was used to determine the ^1^O_2_ production of the PS/NP mix after PDT treatment. A standard curve was constructed with methylene blue from 2.5 to 20 μM in PBS solution. The PS/NP mix at a proportion of 1:32 was used at 2 and 20 μg/mL CuC1. The PDT was performed for 10 min with 61.2 J/cm^2^ of blue LED lamp (450–460 nm). Following the manufacturer’s recommendations, the reaction to determine H_2_O_2_ was performed in a white 96-well plate, adding 10 μL of the H_2_O_2_ substrate to the samples and incubating at room temperature for 60 min. After incubation, 40 μL of the ROS-Glo^TM^ detection solution was added to each well and incubated for 20 min at 20 °C. The luminescence produced was measured in an Infinite M2 pro (Tecan) spectrofluorometer. The reaction to determine the ^1^O_2_ was performed by adding 1 μM SOSG reagent to the samples in a black 96-well plate and measured at excitation/emission, 488/525 nm, in an Infinite M2 pro (Tecan) spectrofluorometer.

### 4.4. Phenotypic and Genotypic Characterization of Sensitivity of the Strains Used

Bacterial cultures were routinely performed in broth or agar Luria–Bertani medium. Axenic cultures were obtained from a single colony of streaks on LB agar plates from stocks maintained at −80 °C. Liquid cultures were grown in LB broth to an OD_600_ of 0.4–0.6 nm for assays. For the phenotypic determination of ESBL-producing strains, the antibiotic susceptibility was carried out using the Kirby–Bauer agar diffusion methodology following the recommendations of the Clinical and Laboratory Standards Institute (CLSI) [46]. Liquid cultures grown were adjusted to a density of 0.5 MacFarland and plated on Muller–Hinton agar. The antibiotic disks were placed radially equidistant for Ceftazidime (30 µg), Ceftazidime–Clavulanic acid (30–10 µg), Ceftriaxone (30 µg), and Ceftriaxone–Clavulanic acid (30–10 µg). For the genotypic characterization of the ESBL-producing strain, the methodology described by Monstein et al., 2007 was performed with the multiplex PCR using the primers listed in Table 4 for detection of the *bla*SHV, *bla*TEM and *bla*CTX-M genes [47]. Total DNA from KPPR-1 and ST258 strains liquid cultures were extracted using the phenol: chloroform methodology and suspended in nuclease-free water. Samples were used for PCR reaction with the GoTaqFlex DNA polymerase kit (Promega) in a final reaction that contained 0.3 µM of each primer, 0.2 µM dNTPs and 3 µM MgCl_2_ in 1x buffer. The reaction was performed in a VeritiFlex (Applied&Biosistem, Foster City, CA, USA) PCR machine, with 1 min at 95 °C for the initial denaturation followed by 30 cycles of 15 s at 95 °C of denaturation, followed by 15 s at 56 °C of annealing and 30 s at 72 °C of extension, with a final extension for 10 min at 72 °C. The PCR products were visualized by electrophoresis fractioning in a 1.5% agar, and the image was captured with an ImageQuant Las500 (General Electric, Boston, MS, USA) photodocumentation system.

### 4.5. Antimicrobial Activity of Photosensitizer Compounds/Nanoparticle Systems

For PDT assays, axenic cultures grown in LB broth to an OD_600_ of 0.4–0.6 nm were adjusted to 1 × 10^7^ colony-forming units (CFU)/mL. Photosensitizer compounds were prepared from a purified powder and dissolved in acetone at 2 mg/mL. The nanoparticles were suspended at 2 mg/mL in ethanol, dispersed by ultrasound and thoroughly vortexed. To prepare the photosensitizer/nanoparticle mix, the nanoparticles suspension was sonicated for 20 min at 40 °C and thoroughly vortexed before taking the required quantity and mixed with the photosensitizers in an aqueous solution just before being used. Photodynamic assays were performed in 24-well plates in a 200 µL final volume in PBS 1x, combining 1 × 10^7^ CFU/mL bacteria with a defined quantity of photosensitizer, nanoparticle, or the mixture of photosensitizer and nanoparticles. The plates were immediately PDT activated for 10 min by a light dose of 61.2 J/cm^2^ in a light chamber at 450–460 nm (with a blue LED lamp, ABI GR-PAR38). The lamp is a 15-chip LED bulb that, at 6 inches (15.24 cm), provides an irradiance of 102 mW/cm^2^ [90].

Bacterial viability was determined by broth microdilution and colony counting on ca-MH agar plates incubated for 16–20 h in the dark at 37 °C. The colony count is expressed as media ±SD of CFU/mL. Control wells with bacteria with no treatment were also included. We established that a photosensitizer effectively reduces bacterial viability by at least 3 log_10_, as suggested by the CLSI [46].

### 4.6. Transcriptional Response to Stress

The transcriptional response to PDT treatment was analyzed by qRT-PCR of total RNA from treated compared to untreated bacteria as we did before [35]. Following the manufacturer protocols, total RNA was extracted using 1 mL of TRI reagent (Thermo Fisher Scientific. 168 Third Avenue. Waltham, MA, USA). The bacterial cells were lysed in TRI reagent during 10 min incubation at room temperature. Cellular debris and genomic DNA were removed by centrifugation at 12,000× *g* at 4 °C for 15 min, and the lysates were recovered in a fresh tube. The lysates were added with 200 µL chloroform and shaken thoroughly for 15 s, and the aqueous phase was recovered by centrifugation. The aqueous phase suspended RNA was precipitated for 10 min with 500 µL of 2-propanol at 4 °C. The RNAs were pelleted by centrifugation for 40 min at 15,000× *g* at 4 °C, washed two times with 80% ethanol at 4 °C and air dried. After drying, the RNAs were resuspended in nuclease-free water and stored at −80 °C. The contaminating DNA was enzymatically removed from the total RNA using 1 U of RNase-free DNase (Promega Corporation, Madison, USA) for 30 min at 37 °C.

The Impront II kit (Promega Corporation, Madison, WI, USA) synthesized CDNA from total RNA in a reverse transcription assay to determine messenger-RNA abundance. cDNA was used to quantify oxidative stress-associated genes relative to the primers in Table 5. The reaction was carried out in a QuantStudio 5 (Applied & Biosystems, Foster City, CA, USA) PCR machine using the Brilliant SYBR^®^ Green QPCR Master Mix (Agilent Technologies, Sta Clara, CA, USA). The relative variation in mRNA abundance of treated samples compared to untreated controls was determined using the ∆∆Ct method.

### 4.7. Cell Culture and Cytotoxicity Tests in Eukaryotic Cells

The human cell lines Hep-G2 and HEK293 were obtained from the American-type culture collection (ATCC) with the catalog codes CCL-23 and CRL-3216, respectively. Cell lines were grown in DMEM with 1% streptomycin/penicillin antibiotics, supplemented with 10% fetal bovine serum (FBS) and incubated at 37 °C in a 5% CO_2_ atmosphere. Initial cultures of 3 × 10^6^ Hep-G2 and 5 × 10^6^ HEK293 cell lines were grown in 10 cm plates for 24 to 48 h to a 70–90% confluence.

The cytotoxic effect of 4 µg/mL CuCl and 64 µg/mL ZnONP, the PS/NP mix, on mammalian tissues was assessed by measuring two parameters of cytotoxicity: cell death and cell survival. Cell death was determined using trypan blue exclusion for the Hep-G2 and HEK-293 cells. For the PDT experiments, cells were prepared from the initial cultures; adherent cells were trypsinized and suspended in a supplemented DMEM medium. Then, 1 × 10^5^ Hep-G2 or 8 × 10^4^ HEK-293 cells were seeded in 24-well plates for 24 to 48 h to achieve 70–90% confluence by incubating at 37 °C in a 5% CO_2_ atmosphere. After incubation, in triplicate, the cell culture medium was removed, washed once with equilibrated D-PBS and then the PS/NP mix in 1x D-PBS was added to the experimental wells. Control wells without PS/NP mix treatment were also included. The plates were incubated for 10 min in the dark or light-activated PS/NP mix (PDT) with a dose of 61.2 J/cm^2^ in a light chamber using a blue 450–460 nm LED lamp. After removing the medium, the adherent cells were washed once with equilibrated D-PBS and incubated in the dark for 24 h with DMEM + 10% FBS in a 5% CO_2_ atmosphere. Following the 24 h incubation, adherent cells were trypsinized and mixed 1:1 with a trypan blue solution. Cell death was determined in a hemocytometer chamber through trypan blue exclusion.

The cell survival was evaluated using the colony-forming assay of cells in vitro [52]. Then, 3 × 10^5^ Hep-G2 or HEK-293 cells were seeded in 6-well plates for 24 to 48 h to achieve 70–90% confluence by incubating at 37 °C in a 5% CO_2_ atmosphere. The semiconfluent cultures were incubated with the PS/NP mix in 1x D-PBS for treatment. One group was kept in the dark (PS/NP), while another was treated for 10 min with 61.2 J/cm^2^ of blue light (PDT). A control group (Crtl) with no PS/NP mix was included. Following treatment, cells were trypsinized with 200 µL Trypsin/EDTA, and an aliquot containing either 500 or 1000 Hep-G2 or 500 KEK293 cells was seeded into a 6-well plate containing DMEM + 10% FBS medium and incubated for 5 to 7 days at 37 °C with 5% CO_2_. Once colonies formed, the medium was removed, and the cells were fixed with 100% methanol for 15 min and stained with 0.5% crystal violet for another 15 min. Colonies consisting of 50 or more cells were recorded, and cell viability was calculated as the number of colonies formed divided by the number of seeded cells, comparing treated versus untreated cells. The plating efficiency (PE) was determined using Formula (1), where the number of control colonies was divided by the number of seeded cells. The cell viability of treated cells was determined as the surviving fraction using Formula (2), where the ratio of the number of colonies to the number of seeded cells was divided by the calculated PE.(1)$$PE\%=\left(\frac{N{}^{\circ}\;\mathrm{colony}\;\mathrm{Control}}{N{}^{\circ}\;\mathrm{Cells}\;\mathrm{seeded}}\right)\times 100$$(2)$$SF=\left(\frac{N{}^{\circ}\;\mathrm{colony}\;\mathrm{Treated}}{N{}^{\circ}\;\mathrm{Cells}\;\mathrm{seeded}\;\times \;\mathrm{PE}}\right)$$

### 4.8. Statistical Analysis

The Prism 6.0 Software (GraphPad Software, LLC, San Diego, CA, USA) was used for statistical analysis. For parametric groups, the statistical significance was determined using the two-tailed *t*-test and one-way ANOVA for the lethality curves accompanied by Tukey’s post-test. The relative quantification was determined using the QuantStudio Design and Analysis v2.7 software (Applied & Biosystems, Foster City, CA 94404 USA).

## 5. Conclusions

PDT can help to cope with the widespread increase in multi-drug-resistant strains, complementing conventional therapies. Although our copper(I)-based compounds have antimicrobial bactericidal activity, this is significantly increased by coupling with nanoparticles. The bactericidal activity of the PS/NP mixture occurs at low concentrations, in ranges like those used in antibiotic therapies (CLSI 2017). Consequently, low-concentration regimes of the PS/NP mixture can be used effectively. Furthermore, the mixture does not show in vitro cytotoxicity to mammalian cells at these concentrations, suggesting its safety in living tissues. In addition, the mode of action is compatible with the Type II effect, which allows the production of prolonged photooxidative stress with the deterioration of the bacterial envelope.

## Figures and Tables

**Figure 1 ijms-26-04178-f001:**
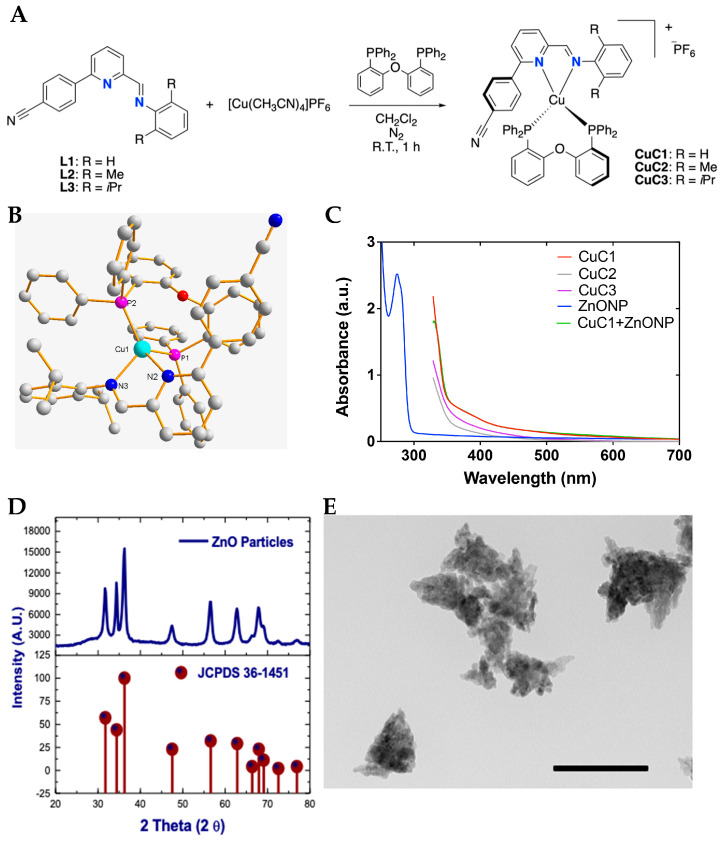
Synthesis and characterization of the photosensitizer and nanoparticles complex. Synthesis of copper(I) complexes CuC1–3 (**A**). Molecular structure of CuC3 obtained by XRD. Thermal ellipsoids of CuC3 are shown to have a 30% probability. Hydrogen atoms and the counter ion were omitted for clarity (**B**). Absorption of CuC1–3 complexes, CuC1 + ZnONP in acetone solution and ZnONP in ethanol solution (**C**). X-ray diffraction of the ZnO obtained from Zincite JC PDF 36–1451 and for synthesized ZnO particles (**D**). TEM images of ZnONP. The dimensions bar represents 200 nm length (**E**).

**Figure 2 ijms-26-04178-f002:**
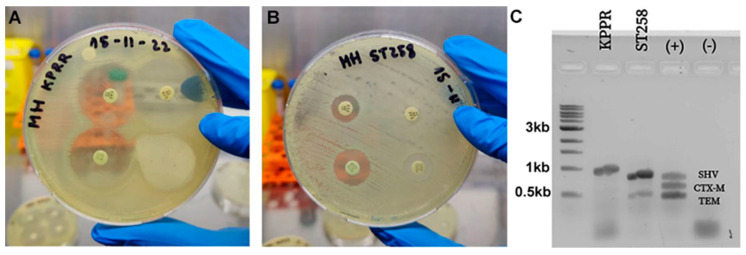
Characterization of ESBL-producing *K. pneumoniae* strains. The *K. pneumoniae* KPPR-1 and ST258 laboratory strains were confirmed to be susceptible and to be extended-spectrum β-lactamase producers, respectively. The Kirby–Bauer diffusion test identified the KPPR-1 *K. pneumoniae* as sensitive to Caz (upper left) and Cro (lower left) and susceptible to the combinations of Caz + Cla (upper right) and Cro + Cla (lower right), where the inhibition halo exceeded 5 mm in each case (**A**). In contrast, the ST258 strain was not sensitive to Caz or Cro (upper right and lower right, respectively) but was sensitive to the combinations of Caz + Cla (upper left) and Cro + Cla (lower left), categorizing it as an ESBL producer (**B**). Through PCR, the genotypes of the KPPR-1 (SHV) and ST258 (SHV and TEM) strains were analyzed and compared to the control strain (SHV, CTX-M, and TEM) (**C**). Both strains carried the SHV resistance gene, which is common to all *K. pneumoniae*; only the TS258 strain contained the TEM gene that encodes resistance to ceftazidime. A control DNA showing three bands was utilized (+), while a negative control (−) without DNA template exhibited only the primer dimer bands.

**Figure 3 ijms-26-04178-f003:**
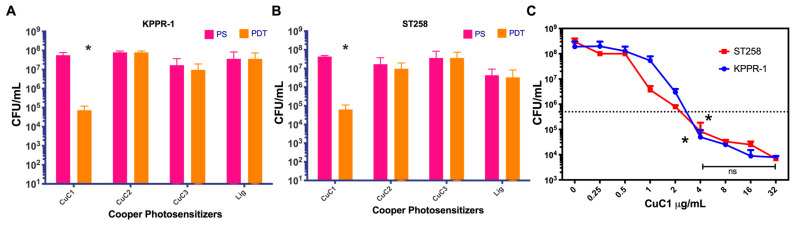
Determination of the photodynamic activity of copper-based photosensitizer compounds. The cooper-based photosensitizers CuC1–C3 photodynamic properties were tested in *K. pneumoniae* KPPR-1 (**A**) and ST258 (**B**) strains. The CuC1 MEC was determined at 0–32 µg/mL (**C**). Bacterial viability is expressed as log_10_ of mean ± SD. ns = *p* > 0.05; * = *p* < 0.05 two-tailed *t*-test compared to control bacteria in the dark.

**Figure 4 ijms-26-04178-f004:**
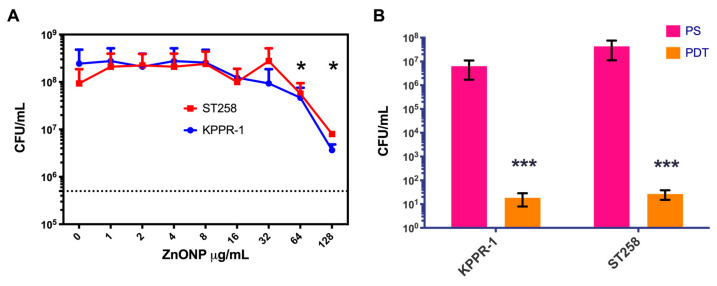
Determination of the photodynamic activity of CuC1 coupled to nanoparticles. The ZnONP MEC was determined in an interval of 0–128 µg/mL and was tested against the KPPR-1 and ST258 *K. pneumoniae* strains (**A**). The photodynamic properties of 4 µg/mL of CuC1 photosensitizer coupled with 80 µg/mL of the zinc oxide nanoparticle (**B**). Bacterial viability is expressed as log_10_ of mean ± SD * = *p* < 0.05; *** = *p* < 0.001 two-tailed *t*-test compared to control bacteria in the dark.

**Figure 5 ijms-26-04178-f005:**
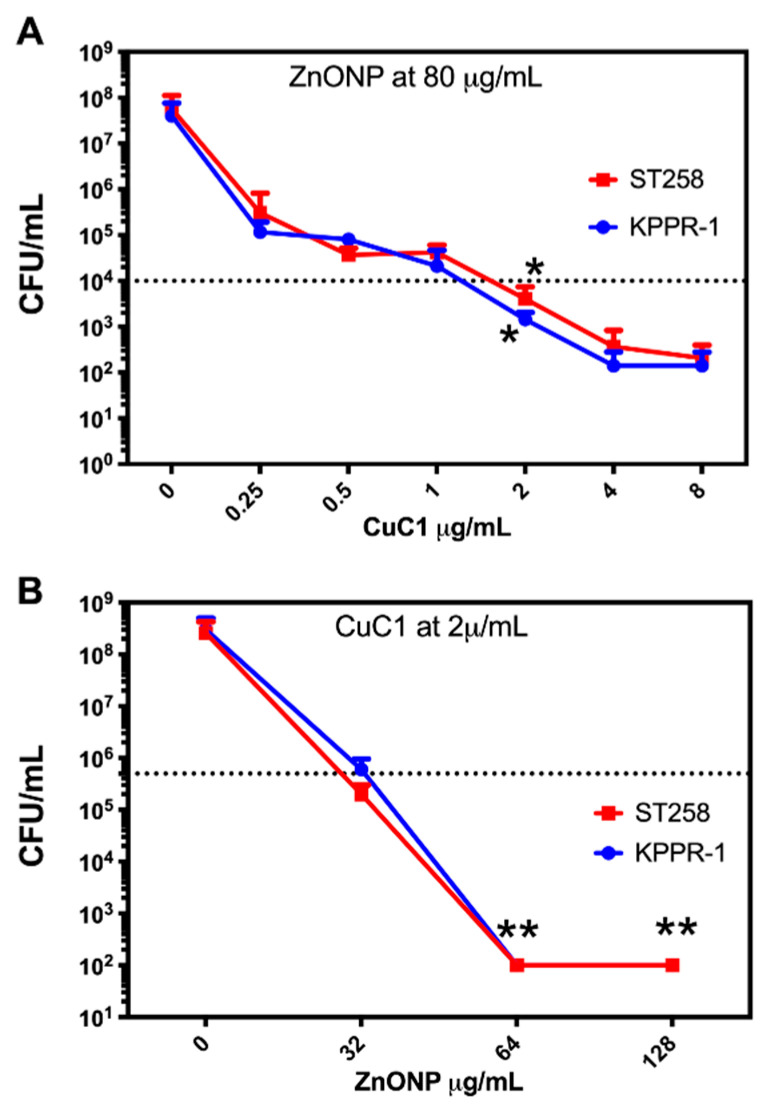
Determination of the minimum inhibitory concentration of CuCl + ZnONP mix. The minimal effective concentration of serial dilutions of CuCl photosensitizer mixed with a fixed concentration of 80 µg/mL of ZnONP was determined in *K. pneumoniae* KPPR-1 and ST258 strains (**A**). The minimal effective concentration of serial dilutions of 0–128 µg/mL of ZnONP photosensitizer mixed with a fixed concentration of 2 µg/mL of CuCl was determined in *K. pneumoniae* KPPR-1 and ST258 strains (**B**). Bacterial viability was assessed after PDT exposure and expressed as log_10_ of mean ± SD * = *p* < 0.05; ** = *p* < 0.01 ANOVA test compared to untreated control.

**Figure 6 ijms-26-04178-f006:**
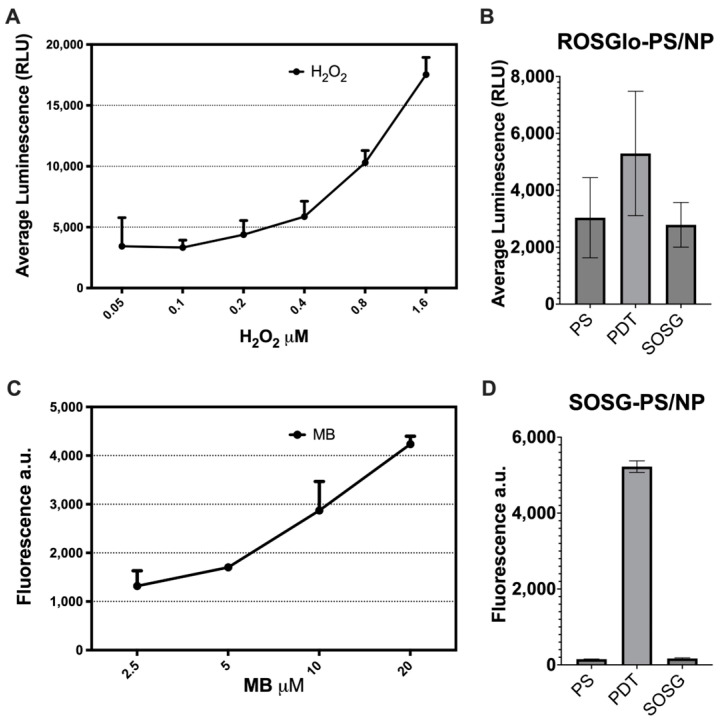
Detection of reactive oxygen species produced by PS/NP mix. Luminescence (RLU) of the H_2_O_2_ (**A**) and the fluorescence of the MB (**C**), standard curves. Luminescence (**B**) and fluorescence (**D**) of the 1:32 PS/NP mix at 2 μg/mL CuC1 activated (PDT) or without light activation (PS) and probe solely without PS/NP mix (ROS-Glo or SOSG, respectively).

**Figure 7 ijms-26-04178-f007:**
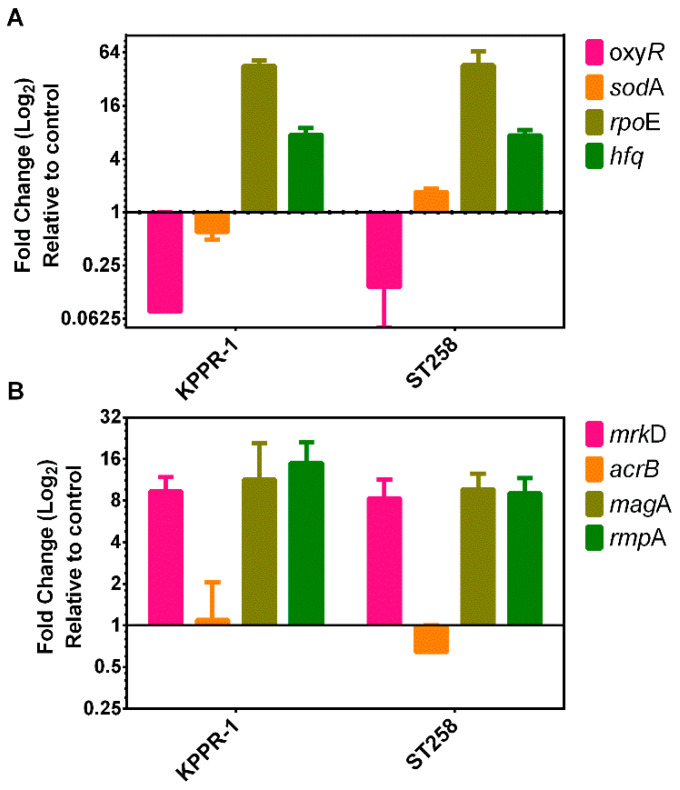
Transcriptional response of *K. pneumoniae* to photooxidative stress. The transcriptional response of *K. pneumoniae* strains KPPR-1 and ST258 to photodynamic stress induced by the CuCl/ZnOPN mixture was determined using RT-qPCR. Changes in the expression of genes related to the oxidative response, including *oxyR*, *sodA*, *rpoE* and *hfq* (**A**), as well as extracytoplasmic genes involved in the restoration of the bacterial envelope, such as *mrkD*, *acrB*, *magA* and *rmpA* (**B**), were assessed. Variations in gene expression were evaluated by analyzing RNA abundance through relative quantification using the 2^−ΔCt^ method compared to the untreated controls.

**Figure 8 ijms-26-04178-f008:**
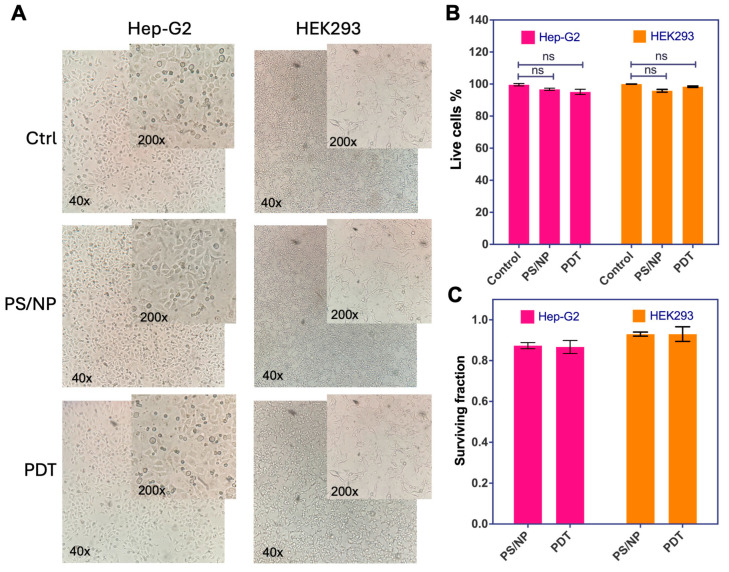
Cytotoxicity in mammalian cells of the CuC1 + ZnONP mixture. The survival of human cell lines Hep-G2 and HEK-293 exposed to a mix of 2 µg/mL CuC1 with 64 µg/mL ZnONP, unexcited (PS/NP), and excited with 61.2 J/cm^2^ of blue light (PDT) was compared to the untreated control cells (Ctrl) (**A**). Cell death was determined by the exclusion of trypan blue and expressed as a percentage of living cells of treated compared to untreated control cells (**B**). Cell viability was determined by colony-forming assay expressed as surviving fraction (**C**). ns = *p* > 0.05, using Student’s *t*-test between treated and untreated cells.

**Table 1 ijms-26-04178-t001:** Effects of PDT with photosensitizer combined with non-biodegradable NPs on bacteria.

Bacteria	Gram Staining	NP as Nanocarrier	Microbial Eradication Rate	References
*S. aureus*	positive	MB-AuNP	95.6%	[27]
*S. aureus*	Positive	Zr/Ti porphyyrinoid metal-organic	99–100%	[28]
*K. pneumoniae*	Negative	MB-AuNP-ConA	97%	[29]

**Table 2 ijms-26-04178-t002:** Photophysical properties of CuC1 and PS control.

Compounds	λ_ab_/nm	λ_em_/nm	Φ_em_
CuC1	380, 421, 460	600	0.038
PSRu-L3	281, 428, 460	608 ^a^	0.156 ^a^

^a^ Extracted from Hormazábal et al., 2023 [35]. λ_ab_ = absorbance wavelength. λ_em_ = emission wavelength. Φ_em_ = emission quantum yield.

**Table 3 ijms-26-04178-t003:** Plating efficiency and surviving fraction.

		Hep-G2	HEK293
	PE	17%	22.3%
SF	PS/NP	0.87 ± 0.015	0.93 ± 0.010
	PDT	0.86 ± 0.032	0.93 ± 0.036

**Table 4 ijms-26-04178-t004:** Primers used for ESBL-production characterization.

Primer Name	Primer Sequence	TM.	Amplicon Size	Origin
bla-SHV.SE	ATGCGTTATATTCGCCTGTG	45	747	[88]
bla-SHV.AS	TGCTTTGTTATTCGGGCCAA	45		
TEM-164.SE	TCGCCGCATACACTATTCTCAGAATGA	53	445	[47]
TEM-165.AS	ACGCTCACCGGCTCCAGATTTAT	52		
CTX-M-U1	ATGTGCAGYACCAGTAARGTKATGGC	54	593	[89]
CTX-M-U2	TGGGTRAARTARGTSACCAGAAYCAGCGG	58		

**Table 5 ijms-26-04178-t005:** List of oligonucleotides used in this paper for qPCR.

Gene	Primers	Gene Type	Origen
*oxyR*	TCCCGAAGCTGGAAATGTAT	Oxidative and nitrosative stress transcriptional regulator	[50]
	GAGCATAATAAGGCGAAAGA		
*sodA*	TTCCGGCTTCCCGATTATCGGCCT	Superoxide dismutase	[50]
	AGCTTCGTCCCAGTTCACTA		
*rpoE*	AACGGGTCCAGAAAGGAGAT	Gene encoding the Sigma factor 32	[50]
	CCTGAACAACGTCAGCGATA		
*Hfq*	ATGGCTAAGGGGCAATCTTT	Posttranscriptional regulation	[50]
	GCTTGATACCATTCACCAAA		
*mrkD*	AAGCTATCGCTGTACTTCCGGCA	Adhesin type 3 fimbriae	[91]
	GGCGTTGGCGCTCAGATAGG		
*magA*	GGTGCTCTTTACATCATTGC	Capsular serotype K1 and hypermucoviscosity phenotype	[92]
	GCAATGGCCATTTGCGTTAG		
*rmpA*	CATAAGAGTATTGGTTGACAG	Regulator of mucoid phenotype A	[93]
	CTTGCATGAGCCATCTTTCA		
*acrB*	GTAAACGTCGTTGGTTAGCC	Acriflavine resistance protein B	[94]
	CTGTATGAGAGCTGGTCGAT		
*16S rRNA*	ATTTGAAGAGGTTGCAAACGAT	Gene encoding the 16S ribosomal RNA	[50]
	TTCACTCTGAAGTTTTCTTGTGTTC		

## Data Availability

The data presented in this study are available on request from the corresponding author. The data are not publicly available because they are confidential data of patients protected by the informed consent protocol.

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
