# Peer review of "Photodynamic Effectiveness of Copper-Iminopyridine Photosensitizers Coupled to Zinc Oxide Nanoparticles Against Klebsiella pneumoniae and the Bacterial Response to Oxidative Stress"

_ijms, 2025, doi:10.3390/ijms26094178_

Round 1
Reviewer 1 Report
Comments and Suggestions for Authors
The submitted manuscript is very interesting and it falls within the scope of the journal. A few suggestions:
- Consider adding a table in the Introduction section summarizing some other materials that have been used in similar applications and highlight the novelty of the present work.
- Please add in the Materials section the purity and origin of the utilised materials for the synthesis.
- Please update the Supplementary materials section with all the information present in the supplementary file.
Author Response
Reviewer 1.
The submitted manuscript is very interesting and it falls within the scope of the journal. A few suggestions:
- Consider adding a table in the Introduction section summarizing some other materials that have been used in similar applications and highlight the novelty of the present work.
Answer: As suggested by the reviewer, the following paragraph was added to the introduction, including a mini-table:
Introduction:
“The impact of PDT using photosensitizers and nanoparticles (NPs) has been investigated in vitro on both Gram-positive and Gram-negative bacteria (Bekmukhametova et al. 2020) (Table 1). Non-biodegradable NPs, such as gold, offer biocompatibility and adjustable sizes for drug delivery (Yeh et al. 2012, Singh et al. 2018). Due to their lack of biological degradation, photosensitizers are released along with the diffusion of singlet oxygen upon light activation (Bekmukhametova et al. 2020). Gold nanoparticles (AuNPs) conjugated with methylene blue (MB-AuNPs) achieved a 96% eradication rate of S. aureus (Tawfik et al. 2015). Zirconium (Zr)-based metal-organic NPs, modified with titanium (Ti) to boost ROS generation, resulted in nearly 100% inhibition of S. aureus (Chen et al. 2020). Dextran-capped MB-AuNPs linked to concanavalin A (Con A) targeted K. pneumoniae, achieving 97% eradication through lectin-mediated adhesion to bacterial cell walls following light activation (Khan et al. 2017).”
Table 1. Effect of PDT with photosensitizer in combination with non-biodegradable NPs on Bacteria.
Bacteria |
Gram staining |
NP as nanocarrier |
Microbial eradication rate |
References |
S. aureus |
positive |
MB-AuNP |
95,6% |
(Tawfik et al. 2015). |
S. aureus |
Positive |
Zr/Ti porphyyrinoid metal-organic |
99-100% |
(Chen et al. 2020) |
K.pneumoniae |
Negative |
MB-AuNP-ConA |
97% |
(Khan et al. 2017) |
- Please add in the Materials section the purity and origin of the utilized materials for the synthesis.
Answer: In response to the reviewer's suggestion, we have added the following sentence to the Materials section (Line 615, Page 16): "All commercially available reagents and solvents were used as received unless otherwise specified. The purity of the reagents is as follows: 4-(6-Formylpyridin-2-yl)benzonitrile (97%), aniline (99%), 2,6-dimethylaniline (98%), 2,6-diisopropylaniline (97%), [Cu(CH3CN)4]BF4 (97%), and DPEphos (98%)."
Regarding commercial origin, we prefer not to specify a trademark since our work was not funded by any companies selling the reagents. With the chemical information provided, the reagent can be acquired from various companies.
- Please update the Supplementary materials section with all the information present in the supplementary file.
Answer: We thank the referee for the comment. After reviewing the information on the physical-chemical characterization of the compounds in both the supplementary materials section and the supplementary file, we can confirm that the chemical information is identical in both documents. The unequivocal assignment of the molecules is found in the supplementary file, where the spectra and the molecular assignment scheme are present. This arrangement makes it much easier to follow the assignment. However, including this level of detail in the manuscript does not contribute further to the discussion and would likely confuse the reader, since the assignment scheme is not included in the main body of the work.
Reviewer 2 Report
Comments and Suggestions for Authors
Minor comments:
(Line 26-39): What is the mechanism by which photodynamic therapy (PDT) reduces bacterial viability in multi-drug resistant (MDR) Klebsiella pneumoniae strains?
(Line 30-32, Line 106-108): How does the combination of copper(I)-based photosensitizer (CuC1) and Zinc Oxide Nanoparticles (ZnONP) enhance phototherapeutic action?
(Line 37-39, Line 295-301): What is the minimum effective concentration (MEC) of CuC1 and ZnONP required for significant bacterial growth inhibition?
(Line 324-341, Line 504-515): What are the predominant reactive oxygen species (ROS) generated by the CuC1/ZnONP photodynamic system?
(Line 367-380, Line 521-525): How does the bacterial transcriptional response differ between Type I and Type II oxidative stress mechanisms in K. pneumoniae?
(Line 140-150, Line 456-459): What structural characteristics of CuC1 contribute to its photodynamic efficiency compared to CuC2 and CuC3?
(Line 100-103, Line 487-490): How do ZnO nanoparticles contribute to increased ROS generation and bacterial cell envelope damage?
(Line 377-380, Line 524-526): What is the role of rpoE and hfq gene upregulation in the bacterial response to photodynamic stress?
(Line 401-417, Line 543-550): What is the potential cytotoxic effect of CuC1/ZnONP on mammalian cell lines, and how does it compare to its antibacterial effects?
(Line 299-302, Line 494-497): Can ZnONP be used at lower concentrations while maintaining photodynamic antimicrobial efficacy?
(Line 313-320, Line 466-471): How does the physical interaction between CuC1 and ZnONP enhance stability and antimicrobial activity?
Author Response
Reviewer 2.
Minor comments:
(Line 26-39): What is the mechanism by which photodynamic therapy (PDT) reduces bacterial viability in multi-drug resistant (MDR) Klebsiella pneumoniae strains?
Answer. We have not investigated the exact mechanism that disrupts the growth of MDR bacteria. However, based on current knowledge and our results, the production of Type II ROS must be responsible. These ROS, particularly singlet oxygen, exert their effects locally and non-specifically, damaging structures of the bacterial cell envelope without distinguishing between sensitive and resistant bacteria.
We have included the following paragraph: “…(PS) that generate reactive oxygen species (ROS) when activated by light. These ROS produce localized oxidative stress damaging the bacterial envelope. A downside of PDT”
(Line 30-32, Line 106-108): How does the combination of copper(I)-based photosensitizer (CuC1) and Zinc Oxide Nanoparticles (ZnONP) enhance phototherapeutic action?
Answer. Although the precise mechanism is still unknown, it is a fact that the bactericidal activity of PS increases when combined with NP. This indicates that in the presence of NP, it is possible to reduce the PS dose to achieve the same effect. This effect results from direct PS–NP interaction, as demonstrated by the binding of the photosensitizer to the nanoparticle.
(Line 37-39, Line 295-301): What is the minimum effective concentration (MEC) of CuC1 and ZnONP required for significant bacterial growth inhibition?
This effect results from direct PS–NP interaction, as demonstrated by the binding of the photosensitizer to the nanoparticle.
Answer: As the reviewer suggested, we have included the MEC doses in the Introduction section. “Combining 2 mg/mL of the photosensitizers with 64 mg/mL of ZnO nanoparticles may improve….” And in the results section, “inhibition. Then, the MEC for the photosensitizer/nanoparticle mixture is 2 mg/mL CuC1 with 64 mg/mL ZnONP.”
(Line 324-341, Line 504-515): What are the predominant reactive oxygen species (ROS) generated by the CuC1/ZnONP photodynamic system?
Answer: As suggested by the reviewer, the experiment's conclusion was included at the end of the paragraph: ….”fluorescence. Thus, the combination of low H₂O₂ production with high ¹O₂ generation indicates that the CuCl/ZnONP photodynamic system primarily induces Type II ROS, establishing it as the main mechanism for ROS generation…”
(Line 367-380, Line 521-525): How does the bacterial transcriptional response differ between Type I and Type II oxidative stress mechanisms in K. pneumoniae?
Answer: As requested by the reviewer, an explanation of the transcriptional response was added to the results. “Consequently, the lack of regulation of the oxyR and sodA genes, coupled with the upregulation of rpoE and hfq, corresponds with the bacterial response to Type II photooxidative stress. This confirms that singlet oxygen production is the primary reactive oxygen species (ROS) generated by CuCl/ZnONP photodynamic therapy.”
(Line 140-150, Line 456-459): What structural characteristics of CuC1 contribute to its photodynamic efficiency compared to CuC2 and CuC3?
Answer: We thank the reviewer for his observation and have added the following phrase in the Discussion section of the manuscript (Line 499, Page 14): “This activity may be influenced by steric hindrance around the metal center. The CuC1 complex experiences the least steric hindrance, featuring a hydrogen substituent, while CuC3 displays the greatest steric effect due to the presence of isopropyl groups on the aromatic ring. The ability of the complex to undergo geometric changes during Cu(I)/Cu(II) interconversion is determined by the structural rigidity around the metal center.[10.1039/B906013H] Complexes with lower structural rigidity experience less resistance to geometric change, leading to lower redox potentials, which directly influence the single-electron transfer.[10.1039/C4DT02046D] Furthermore, increasing the steric bulk of the ligand around the metal center may impact the efficiency of single-electron transfer.[10.1007/s11243-018-0237-1]”
(Line 100-103, Line 487-490): How do ZnO nanoparticles contribute to increased ROS generation and bacterial cell envelope damage?
Answer: as the reviewer suggests, a paragraph was added in lines 110 – 121.
(Line 377-380, Line 524-526): What is the role of rpoE and hfq gene upregulation in the bacterial response to photodynamic stress?
Answer: As suggested by the reviewer, an explanatory text was added in line 390: “…(Figure 7A). In the R. sphaeroides bacteria, the extra-cytoplasmatic factor rpoE, as well as the hfq post-transcriptional regulator genes, are overexpressed upon 1O2 stressor. RpoE controls several genes that specifically respond to 1O2 stress and are posttranscriptional modulated by the Hfq factor. Like R. sphaeroides…”
(Line 401-417, Line 543-550): What is the potential cytotoxic effect of CuC1/ZnONP on mammalian cell lines, and how does it compare to its antibacterial effects?
Answer: As suggested by the reviewer, we have added the following explanatory phrases to the discussion section to address potential differences in the effects of PDT on mammalian and bacterial cells.
“Finally, we needed to evaluate the safety of the compound by analyzing evidence of cytotoxicity in mammalian cells because PDT induces nonspecific photooxidative stress, which can disrupt the normal metabolism of eukaryotic host cells. As previously observed, mammalian cells tolerate the exposure to 1O2 well, led by the PS/ZnoNP mix, as we observe very low or absent compound-induced cytotoxicity. We used hepatocyte-derived Hep-G2 cell lines from human sources to assess any potential liver toxicity, a crucial organ for drug detoxification (Zheng et al. 2024). We also did not see cytotoxicity in human kidney-derived HEK-293 cells, confirming the absence of a potential renal effect that could explain impaired drug elimination (Costa et al. 2024). Given that the antimicrobial activity of PDT occurs immediately after light exposure, we replicated the same conditions for the eukaryotic cells. Images of cultured cells show no evidence of cytopathic damage, nor does there appear to be any loss of cell viability in trypan blue exclusion assays or colony formation assays. Therefore, the bactericidal effect of photooxidative stress, through the generation of 1O2, does not affect mammalian cells.”
(Line 299-302, Line 494-497): Can ZnONP be used at lower concentrations while maintaining photodynamic antimicrobial efficacy?
Answer: We observed that the better ratio is 1:32, and the photosensitizer CIM in NP presence is 2 ug/mL; then, at least 64 ug/mL of nanoparticles must be used.
(Line 313-320, Line 466-471): How does the physical interaction between CuC1 and ZnONP enhance stability and antimicrobial activity?
Answer: Following the reviewer’s concern, we have added the following phrase in the Discussion section of the manuscript (Line 544, Page 15 ): “The interaction between CuC1 and ZnONPs leads to enhanced stability and antimicrobial activity through multiple physicochemical and biological mechanisms. In terms of stability, ZnONPs function as a protective scaffold that prevents CuC1 from aggregating or degrading. This occurs through electrostatic interactions, hydrogen bonding, or coordination between CuC1 and the hydroxyl-rich surface of ZnONPs (Raju et al. 2022). By altering the electronic environment of CuC1, ZnONPs can stabilize their oxidation states (Cu(I)/Cu(II)), which is crucial for maintaining their redox activity. Beyond stability, the synergistic antimicrobial activity of the CuC1-ZnONP system is primarily driven by enhanced reactive oxygen species (ROS) generation and bacterial membrane disruption. Their combination may further improve photo-Fenton-like reactions, leading to more efficient and sustained ROS production (Vibornijs et al. 2023). Together, these factors render CuC1-ZnONP composites highly effective and durable antimicrobial agents (Yang et al. 2022).
”
Reviewer 3 Report
Comments and Suggestions for Authors
This report discusses the potential of PDT for treatment of bacterial infections that respond poorly to other forms of treatment. The report discusses metalloporphyrins, but insertion of certain metals into the ring can quench photodynamic activity. These metals include Mn and Cu. Absorbance profiles shown in Fig. 1 panel E indicate no values > 400 nm. Emission spectra are irrelevant.
In vitro studies involve a very high light dose: 61.2 J\sq cm (line 239). Most in vitro studies involve orders of magnitude less light. It is not clear how effects on mammalian cells were examined. No clonogenic studies are described. It is not clear how bacterial infections could be treated by the process outlined here. Bacterial infections tend to be systemic: where would light be directed? Trypan blue exclusion is not an adequate test for clonogenic abilty of mammalian cells. Moreover, insertion of heavy metals can quench the photodynamic activity of photosensitizers. It is not clear that photodynanic activity is involved.
What is not clear: how could systemic bacterial infections be treated in vivo? Why are such high light doses needed in vitro? Light doses and light flux are often confused. How can the need for light be overcome (line 490)? There is no information in Fig. 8 concerning the light dose. TB exclusion is, as noted above, not a test for loss of viability. The magnification used in panel A of Fig. 8 is too low to detect any cell damage. The labeling of the Y axis in panel B is misleading since survival is never measured.
Author Response
Reviewer 3.
This report discusses the potential of PDT for treatment of bacterial infections that respond poorly to other forms of treatment. The report discusses metalloporphyrins, but insertion of certain metals into the ring can quench photodynamic activity. These metals include Mn and Cu. Moreover, insertion of heavy metals can quench the photodynamic activity of photosensitizers.
Absorbance profiles shown in Fig. 1 panel E indicate no values > 400 nm. Emission spectra are irrelevant.
Answer: We acknowledge the referee's concern regarding the apparent lack of absorption of these systems at wavelengths greater than 400 nm. However, when analyzing the curves on a larger scale, we observed that the complexes exhibit a shoulder near 400 nm and a tail that decays beyond this wavelength, reaching values between 421 and 460 nm. This point is already addressed in the results section of the manuscript (page 4, lines 169-172): " Additionally, a tail extending to longer wavelengths can be ascribed to spin-allowed metal-to-ligand charge transfer (¹MLCT) transitions [49, 50]. As shown in Table 2, the photosensitizer compounds show absorption bands close to 421 and 460 nm."
On the other hand, based on previous work by members of the research group (Eur. J. Inorg. Chem. 2021, 4020–4029), we can assert that even if a compound does not show an absorption maximum at the same wavelength as the irradiation lamp, it can be excited and generate reactive species. This is because the irradiation wavelength falls within the absorption bandwidth of the complex, regardless of the position of the curve maximum, and has sufficient energy to promote the electronic transition. This was observed in our previous work, both experimentally and theoretically.
Regarding the emission spectra concern, and following the reviewer’s suggestion, we have erased the emission spectra from Figure 1, which can now be found in the Supplementary Material (Figure S17).
In vitro studies involve a very high light dose: 61.2 J\sq cm (line 239). Most in vitro studies involve orders of magnitude less light.
Answer: We appreciate the reviewer's concern. We ran the calculations again, using the lamp manufacturer's declared power, and the results were correct. Furthermore, the fluence of 61.2 J/cm2 is within the range of many results reported by several researchers. Without going any further, we conducted a review ourselves, where values range from 5 to 280 J/cm2, with an average of 120 J/cm2.
https://pmc.ncbi.nlm.nih.gov/articles/PMC11676773/pdf/pharmaceutics-16-01626.pdf.(Bravo et al. 2024).
It is not clear how effects on mammalian cells were examined. No clonogenic studies are described. Trypan blue exclusion is not an adequate test for clonogenic abilty of mammalian cells.
Answer: As suggested by the reviewer, the clonogenic assays were performed for HEK-293 cell lines.
It is not clear how bacterial infections could be treated by the process outlined here. Bacterial infections tend to be systemic: where would light be directed? It is not clear that photodynanic activity is involved.
Answer: There are many ways to apply antimicrobial photodynamic therapy (PDT) in a clinical setting, similar to anticancer PDT. For instance, it is effective for superficial infections, which are very common in surgical wounds caused by K. pneumoniae. PDT is also beneficial when the infection is confined to organs, especially hollow organs, where probes can illuminate the internal walls, as seen in urinary tract infections, a primary type of infection caused by K. pneumoniae. Clearly, antimicrobial PDT is not intended for treating bloodstream infections, but it can help eliminate the source of the infection.
This is explained in lines 535–538, page 15: “…light. The need for light activation can be overcome during the treatment of pneumonia or urinary tract infections by activating the PS/NP mix on-site using devices, such as optical fibers used through a catheter, that deliver light into internal hollow organs, including non-spherical cavities (Svanberg et al. 2010, Baran et al. 2014, Tan et al. 2021).”
What is not clear: how could systemic bacterial infections be treated in vivo? Why are such high light doses needed in vitro?
Answer: This is explained in lines 538–540, page 15: “…While this therapy cannot treat bacteremia or bloodstream infections, it can significantly assist in eliminating the source of the infection, such as a UTI.”
Light doses and light flux are often confused.
Answer: as the reviewer suggested, the use of flux or doses was revised through the text.
How can the need for light be overcome (line 490)?
Answer: the question was answered above.
There is no information in Fig. 8 concerning the light dose.
Answer: … and excited with 61.2 J/cm2 of blue light (PDT)
TB exclusion is, as noted above, not a test for loss of viability. The magnification used in panel A of Fig. 8 is too low to detect any cell damage. The labeling of the Y axis in panel B is misleading since survival is never measured.
Answer: As suggested by the reviewer, the colony-forming assay was performed, and the 200x magnification of the photograph was included. Also, the Y label was changed.
References.
Baran, T. M. and T. H. Foster (2014). "Comparison of flat cleaved and cylindrical diffusing fibers as treatment sources for interstitial photodynamic therapy." Med Phys 41(2): 022701.
Bekmukhametova, A., H. Ruprai, J. M. Hook, D. Mawad, J. Houang and A. Lauto (2020). "Photodynamic therapy with nanoparticles to combat microbial infection and resistance." Nanoscale 12(41): 21034-21059.
Bravo, A. R., F. A. Fuentealba, I. A. Gonzalez and C. E. Palavecino (2024). "Use of Antimicrobial Photodynamic Therapy to Inactivate Multidrug-Resistant Klebsiella pneumoniae: Scoping Review." Pharmaceutics 16(12).
Chen, M., Z. Long, R. Dong, L. Wang, J. Zhang, S. Li, X. Zhao, X. Hou, H. Shao and X. Jiang (2020). "Titanium Incorporation into Zr-Porphyrinic Metal-Organic Frameworks with Enhanced Antibacterial Activity against Multidrug-Resistant Pathogens." Small 16(7): e1906240.
Costa, R. M., M. C. Dias, J. V. Alves, J. L. M. Silva, D. Rodrigues, J. F. Silva, H. D. Francescato, L. N. Ramalho, T. M. Coimbra and R. C. Tostes (2024). "Pharmacological activation of nuclear factor erythroid 2-related factor-2 prevents hyperglycemia-induced renal oxidative damage: Possible involvement of O-GlcNAcylation." Biochemical Pharmacology 220: 115982.
Khan, S., S. N. Khan, R. Meena, A. M. Dar, R. Pal and A. U. Khan (2017). "Photoinactivation of multidrug resistant bacteria by monomeric methylene blue conjugated gold nanoparticles." J Photochem Photobiol B 174: 150-161.
Raju, P., D. Deivatamil, J. A. Martin Mark and J. P. Jesuraj (2022). "Antibacterial and catalytic activity of Cu doped ZnO nanoparticles: structural, optical, and morphological study." Journal of the Iranian Chemical Society 19(3): 861-872.
Singh, P., S. Pandit, V. Mokkapati, A. Garg, V. Ravikumar and I. Mijakovic (2018). "Gold Nanoparticles in Diagnostics and Therapeutics for Human Cancer." Int J Mol Sci 19(7).
Svanberg, K., N. Bendsoe, J. Axelsson, S. Andersson-Engels and S. Svanberg (2010). "Photodynamic therapy: superficial and interstitial illumination." J Biomed Opt 15(4): 041502.
Tan, Y., S. Sun, D. Chen, H. Qiu, J. Zeng, Y. Wang, H. Zhao and Y. Gu (2021). "Light delivery device modelling for homogenous irradiation distribution in photodynamic therapy of non-spherical hollow organs." Photodiagnosis Photodyn Ther 34: 102320.
Tawfik, A. A., J. Alsharnoubi and M. Morsy (2015). "Photodynamic antibacterial enhanced effect of methylene blue-gold nanoparticles conjugate on Staphylococcal aureus isolated from impetigo lesions in vitro study." Photodiagnosis Photodyn Ther 12(2): 215-220.
Vibornijs, V., M. Zubkins, E. Strods, Z. Rudevica, K. Korotkaja, A. Ogurcovs, K. Kundzins, J. Purans and A. Zajakina (2023). "Analysis of antibacterial and antiviral properties of ZnO and Cu coatings deposited by magnetron sputtering: Evaluation of cell viability and ROS production." Coatings 14(1): 14.
Yang, F., Y. Song, A. Hui, B. Mu and A. Wang (2022). "Phyto-mediated controllable synthesis of ZnO clusters with bactericidal activity." ACS Applied Bio Materials 6(1): 277-287.
Yeh, Y. C., B. Creran and V. M. Rotello (2012). "Gold nanoparticles: preparation, properties, and applications in bionanotechnology." Nanoscale 4(6): 1871-1880.
Zheng, J., S. Jung, J.-H. Ha and Y. Jeong (2024). "Locusta migratoria hydrolysates attenuate lipopolysaccharide (LPS)/D-Galactosamine (D-Gal)-induced cytotoxicity and inflammation in Hep G2 cells via NF-κB signaling suppression." Applied Biological Chemistry 67(1): 49.
Round 2
Reviewer 3 Report
Comments and Suggestions for Authors
This report discusses the potential of PDT for treatment of bacterial infections that respond poorly to other forms of treatment. The report discusses metalloporphyrins, but insertion of certain metals into the ring can quench photodynamic activity. These metals include Mn and Cu. Absorbance profiles shown in Fig. 1 panel E indicate no values > 400 nm. Emission spectra are irrelevant.
In vitro studies involve a very high light dose: 61.2 J\sq cm (line 239). Most in vitro studies involve orders of magnitude less light. It is not clear how effects on mammalian cells were examined. No clonogenic studies are described. It is not clear how bacterial infections could be treated by the process outlined here. Bacterial infections tend to be systemic: where would light be directed? Trypan blue exclusion is not an adequate test for clonogenic ability of mammalian cells. Moreover, insertion of heavy metals can quench the photodynamic activity of photosensitizers. It is not clear that photodynamic activity is involved.
Very high light doses are indicated for an in vitro study. Light doses and light flux are often confused. How can the need for light be overcome (line 490)? There is no information in Fig. 8 concerning the light dose. TB exclusion is used as a test for loss of viability. The magnification used in panel A of Fig. 8 is too low to detect any cell damage. The labeling of the Y axis in panel B is misleading since survival is never measured. All of these issues need to be dealt with before this is suitable for publication.
Author Response
Reviewer 3.
This report discusses the potential of PDT for treatment of bacterial infections that respond poorly to other forms of treatment. The report discusses metalloporphyrins, but insertion of certain metals into the ring can quench photodynamic activity. These metals include Mn and Cu. Moreover, insertion of heavy metals can quench the photodynamic activity of photosensitizers.
Absorbance profiles shown in Fig. 1 panel E indicate no values > 400 nm. Emission spectra are irrelevant.
Answer: We acknowledge the referee's concern regarding the apparent lack of absorption of these systems at wavelengths greater than 400 nm. However, when analyzing the curves on a larger scale, we observed that the complexes exhibit a shoulder near 400 nm and a tail that decays beyond this wavelength, reaching values between 421 and 460 nm. This point is already addressed in the results section of the manuscript (page 4, lines 169-172): " Additionally, a tail extending to longer wavelengths can be ascribed to spin-allowed metal-to-ligand charge transfer (¹MLCT) transitions [49, 50]. As shown in Table 2, the photosensitizer compounds show absorption bands close to 421 and 460 nm."
On the other hand, based on previous work by members of the research group (Eur. J. Inorg. Chem. 2021, 4020–4029), we can assert that even if a compound does not show an absorption maximum at the same wavelength as the irradiation lamp, it can be excited and generate reactive species. This is because the irradiation wavelength falls within the absorption bandwidth of the complex, regardless of the position of the curve maximum, and has sufficient energy to promote the electronic transition. This was observed in our previous work, both experimentally and theoretically.
Regarding the emission spectra concern, and following the reviewer’s suggestion, we have erased the emission spectra from Figure 1, which can now be found in the Supplementary Material (Figure S17).
In vitro studies involve a very high light dose: 61.2 J\sq cm (line 239). Most in vitro studies involve orders of magnitude less light.
Answer: We appreciate the reviewer's concern. We ran the calculations again, using the lamp manufacturer's declared power, and the results were correct. Furthermore, the fluence of 61.2 J/cm2 is within the range of many results reported by several researchers. Without going any further, we conducted a review ourselves, where values range from 5 to 280 J/cm2, with an average of 120 J/cm2.
https://pmc.ncbi.nlm.nih.gov/articles/PMC11676773/pdf/pharmaceutics-16-01626.pdf.(Bravo et al. 2024).
It is not clear how effects on mammalian cells were examined. No clonogenic studies are described. Trypan blue exclusion is not an adequate test for clonogenic abilty of mammalian cells.
Answer: As suggested by the reviewer, the clonogenic assays were performed for Hep2-G6 and HEK-293 cell lines.
It is not clear how bacterial infections could be treated by the process outlined here. Bacterial infections tend to be systemic: where would light be directed? It is not clear that photodynanic activity is involved.
Answer: There are many ways to apply antimicrobial photodynamic therapy (PDT) in a clinical setting, similar to anticancer PDT. For instance, it is effective for superficial infections, which are very common in surgical wounds caused by K. pneumoniae. PDT is also beneficial when the infection is confined to organs, especially hollow organs, where probes can illuminate the internal walls, as seen in urinary tract infections, a primary type of infection caused by K. pneumoniae. Clearly, antimicrobial PDT is not intended for treating bloodstream infections, but it can help eliminate the source of the infection.
This is explained in lines 535–538, page 15: “…light. The need for light activation can be overcome during the treatment of pneumonia or urinary tract infections by activating the PS/NP mix on-site using devices, such as optical fibers used through a catheter, that deliver light into internal hollow organs, including non-spherical cavities (Svanberg et al. 2010, Baran et al. 2014, Tan et al. 2021).”
What is not clear: how could systemic bacterial infections be treated in vivo? Why are such high light doses needed in vitro?
Answer: This is explained in lines 538–540, page 15: “…While this therapy cannot treat bacteremia or bloodstream infections, it can significantly assist in eliminating the source of the infection, such as a UTI.”
Light doses and light flux are often confused.
Answer: as the reviewer suggested, the use of flux or doses was revised through the text.
How can the need for light be overcome (line 490)?
Answer: the question was answered above.
There is no information in Fig. 8 concerning the light dose.
Answer: … and excited with 61.2 J/cm2 of blue light (PDT)
TB exclusion is, as noted above, not a test for loss of viability. The magnification used in panel A of Fig. 8 is too low to detect any cell damage. The labeling of the Y axis in panel B is misleading since survival is never measured.
Answer: As suggested by the reviewer, the colony-forming assay was performed, and the 200x magnification of the photograph was included. Also, the Y label was changed.

Round 3
Reviewer 3 Report
Comments and Suggestions for Authors
Since the presence of Cu will quench ROS formation, is it not clear why this element would be incorporated into a photosensitizer. Furthermore, short wavelength irradiation is readily scattered and would have a very limited ability to penetrate into tissues. Microbial infections tend to be systemic. PDT effects are limited to regions within the field of irradiation. Research directed toward metals cited in line 91, i.e., Mn, Fe, or Cu, is unexplained since the presence of these metals will quench ROS formation. ‘High1O2 formation’ is not associated with the presence of any of these metals.
Panel C of Fig. 1 shows that the photosensitizer being examined has no significant absorbance at wavelengths > 400 nm. Very high light doses are needed for efficacy (section 2.3). A dose of 61.2 J/sq cm is very high for in vitro studies; effective photokilling in vitro is usually observe with orders of magnitude less light.
Since PDT can readily eradicate mammalian cell types, the lack of efficacy against Hep-G2 or HEK293 cells remains to be explained. The legend to Fig. 8 indicates that mammalian cells were exposed to CuC1 and ZnONP. Since photosensitized mammalian cells can readily be eradicated by ROS, any preferential efficacy on bacteria might be explained by uptake differences. No studies were carried out to determine whether these photosensitizers were accumulated by mammalian cells vs. microorganisms.
Hydrogen peroxide is generally not considered to be one of the more lethal ROS formed during PDT. What does ‘exposed to PDT’ mean (line 346)? Presumably, this means ‘exposed to light’. In some cases, e.g., as indicated in lines 331 and 718, formation of ROS was detected ‘after PDT’ which presumably means after irradiation. Since reactive oxygen species will promptly react with any nearby oxidizable species, detection reagents should be present during irradiation, not after. It is therefore unclear how ROS determinations were carried out.
What needs to be done: [1] measure uptake of the photosensitizer(s) by mammalian vs. bacterial cell populations; [2] insure that ROS detection occurs with reagents present during rather than after irradiation; [3] explain what microbial infections could be treated by a protocol that requires light for efficacy.
In their commentary, the authors claim that efficacy of PDT may be observed where this no significant light absorbance. This factor does not account for the need for such a light light flux at wavelengths that are clearly at values where there is a significant absorption optimum. Since everything in this report is ‘in vitro’, even higher light doses will likely be required in vivo.
Author Response
Since the presence of Cu will quench ROS formation, is it not clear why this element would be incorporated into a photosensitizer. Furthermore, short wavelength irradiation is readily scattered and would have a very limited ability to penetrate into tissues. Microbial infections tend to be systemic. PDT effects are limited to regions within the field of irradiation. Research directed toward metals cited in line 91, i.e., Mn, Fe, or Cu, is unexplained since the presence of these metals will quench ROS formation. ‘High1O2 formation’ is not associated with the presence of any of these metals.
Answer: While it is well known that certain metal ions, such as Cu(I) and Cu(II), can quench singlet oxygen (1O2) under specific conditions, this property does not universally disqualify copper from being incorporated into effective photosensitizers. In fact, a substantial amount of evidence shows that copper-based coordination complexes can actively participate in ROS generation through alternative pathways, particularly via redox cycling or Fenton-like processes, leading to the formation of hydroxyl radicals and other reactive species. These mechanisms fall under the Type I photodynamic pathway, which is particularly relevant under hypoxic or biologically reducing conditions where the formation of 1O2 is limited (Krasnovskaya et al. 2020, Rani et al. 2023, Da Costa Ferreira et al. 2024).
Moreover, the ligand environment surrounding the Cu center plays a crucial role in modulating its photophysical and redox behavior, enabling the suppression of non-radiative decay and enhancing photoinduced charge separation (Engl et al. 2022, Da Costa Ferreira et al. 2024). Well-designed Cu(I) complexes with diimine or polypyridyl ligands, for instance, have demonstrated high photoinduced cytotoxicity and efficient ROS production in vitro, despite copper’s inherent quenching capability (Mucha et al. 2019, Molinaro et al. 2020, Rani et al. 2023). Furthermore, high production of ¹O₂ has already been detected experimentally in our system (SOSG), which validates our assumptions.
Regarding the issue of short-wavelength irradiation and tissue penetration, we agree that shorter-wavelength light has limited tissue penetration. However, photodynamic therapy is not limited to systemic infections; it is particularly effective for localized or superficial microbial infections, such as those found in wounds, diabetic ulcers, dental biofilms, or skin infections (Allison et al. 2004). Hence, our strategy focuses on these rather than deep systemic infections. This is already outlined in the manuscript, lines 536–542: “The need for light activation can be sustained when treating superficial infections such as surgical wounds (K. pneumoniae is one of the main causes) or other skin infections. For pneumonia or urinary tract infections (UTI), the PS/NP mix can be activated on-site using devices such as optical fibers delivered through a catheter, which transmit light into internal hollow organs, including non-spherical cavities (Svanberg et al. 2010, Baran et al. 2014, Tan et al. 2021). While this therapy cannot treat bacteremia or bloodstream infections, it can significantly assist in eliminating the source of the infection, such as a UTI.”
Panel C of Fig. 1 shows that the photosensitizer being examined has no significant absorbance at wavelengths > 400 nm. Very high light doses are needed for efficacy (section 2.3). A dose of 61.2 J/sq cm is very high for in vitro studies; effective photokilling in vitro is usually observe with orders of magnitude less light.
Answer: Whereas it is correct that Panel C of Fig. 1 indicates low absorbance beyond 400 nm, it is important to note that the photosensitizer still absorbs within the violet-blue region of the visible spectrum, which is sufficient to promote excitation under controlled conditions. Furthermore, using shorter excitation wavelengths—although suboptimal for tissue penetration—is valid for in vitro proof-of-concept studies, where light delivery is not a limiting factor. In this context, our observed biological response confirms that energy transfer to oxygen and/or ROS generation still occurs despite the limited red-shifted absorbance.
Regarding the light dose, while 61.2 J/cm² is higher than what is typically used in some in vitro photokilling assays, it falls within the range of values reported in many studies and was intentionally chosen to ensure maximum activation under relatively low extinction coefficients, which is common for certain transition metal complexes. Furthermore, other studies have utilized even higher light doses to demonstrate photoactivity (Photodiagnosis and Photodynamic Therapy 2023, 44, 103865). Additionally, we have included a review specifically on K. pneumoniae in the manuscript.
Since PDT can readily eradicate mammalian cell types, the lack of efficacy against Hep-G2 or HEK293 cells remains to be explained. The legend to Fig. 8 indicates that mammalian cells were exposed to CuC1 and ZnONP. Since photosensitized mammalian cells can readily be eradicated by ROS, any preferential efficacy on bacteria might be explained by uptake differences. No studies were carried out to determine whether these photosensitizers were accumulated by mammalian cells vs. microorganisms.
Answer: In the manuscript, we explain why these compounds are active against bacteria but not against mammalian cells. The type of ROS produced may account for this. Our PS/NP mix primarily generates singlet oxygen, which enterobacteria seem to lack appropriate response mechanisms for, unlike mammalian cells. The following paragraph appears in lines 611–621: “Both Type I reactive oxygen species (ROS), like the hydroxyl radical (OH⋅), and Type II ROS, such as singlet oxygen (1O2), are highly reactive entities that can cause significant damage to cells. However, the OH⋅ is generally considered more toxic to both mammalian cells and bacteria (Edge et al. 2021). For mammalian cells, OH⋅ is one of the most reactive ROS; even in small amounts, it may cause extensive damage to lipids, proteins, and DNA, resulting in immediate apoptosis or necrosis. Although 1O2 is toxic for both, mammalian cells have more robust antioxidant systems (e.g., superoxide dismutase, catalase, glutathione) that help them mitigate the damage (Szewczyk et al. 2024); then, it can be managed by eukaryotic cellular antioxidant defenses (Jomova et al. 2023). Because the 1O2 tends to react more with lipids and proteins, it can be highly toxic for bacteria, causing oxidative damage to the bacterial envelope components such as the cell membrane, capsule proteins, and also DNA (Kanofsky 1993).”
Hydrogen peroxide is generally not considered to be one of the more lethal ROS formed during PDT. What does ‘exposed to PDT’ mean (line 346)? Presumably, this means ‘exposed to light’. In some cases, e.g., as indicated in lines 331 and 718, formation of ROS was detected ‘after PDT’ which presumably means after irradiation. Since reactive oxygen species will promptly react with any nearby oxidizable species, detection reagents should be present during irradiation, not after. It is therefore unclear how ROS determinations were carried out.
Answer: The method of conducting the experiment has been corrected in lines 338 – 339 and line 727.
What needs to be done: [1] measure uptake of the photosensitizer(s) by mammalian vs. bacterial cell populations; [2] insure that ROS detection occurs with reagents present during rather than after irradiation; [3] explain what microbial infections could be treated by a protocol that requires light for efficacy.
Answer: We have provided a documented explanation of the differential activity of Type I and Type II ROS in mammalian and bacterial cells (lines 611- 621). Measuring the uptake of photosensitizer(s) by mammalian versus bacterial cell populations is beyond the scope of this work. At this point, it may be sufficient to demonstrate the safety of the bactericidal mixture for human cells.
- ROS detection was updated as above (in lines 338 – 339 and line 727).
- Microbial infection, where this technology is useful, was discussed above (lines 536–542).
In their commentary, the authors claim that efficacy of PDT may be observed where this no significant light absorbance. This factor does not account for the need for such a light light flux at wavelengths that are clearly at values where there is a significant absorption optimum. Since everything in this report is ‘in vitro’, even higher light doses will likely be required in vivo.
Answer: We appreciate the reviewer’s insightful comment regarding the use of light doses and the potential need for higher energy input in in-vivo applications, stemming from tissue scattering and limited penetration depth. We fully agree that these factors are critical in translating photodynamic strategies from in-vitro to in-vivo contexts. In this regard, we would like to emphasize that one of our planned future directions involves assessing the safety and efficacy of the reported copper-based complexes in animal models. These studies will allow us to optimize irradiation parameters under physiologically relevant conditions and further validate the therapeutic potential of our system.
References.
Allison, R. R., G. H. Downie, R. Cuenca, X. H. Hu, C. J. Childs and C. H. Sibata (2004). "Photosensitizers in clinical PDT." Photodiagnosis Photodyn Ther 1(1): 27-42.
Baran, T. M. and T. H. Foster (2014). "Comparison of flat cleaved and cylindrical diffusing fibers as treatment sources for interstitial photodynamic therapy." Med Phys 41(2): 022701.
Da Costa Ferreira, A. M., C. Hureau and G. Facchin (2024). "Bioinorganic Chemistry of Copper: From Biochemistry to Pharmacology." Inorganics 12(4): 97.
Edge, R. and T. G. Truscott (2021). "The Reactive Oxygen Species Singlet Oxygen, Hydroxy Radicals, and the Superoxide Radical Anion—Examples of Their Roles in Biology and Medicine." Oxygen 1(2): 77-95.
Engl, S. and O. Reiser (2022). "Copper-photocatalyzed ATRA reactions: concepts, applications, and opportunities." Chemical Society Reviews 51(13): 5287-5299.
Jomova, K., R. Raptova, S. Y. Alomar, S. H. Alwasel, E. Nepovimova, K. Kuca and M. Valko (2023). "Reactive oxygen species, toxicity, oxidative stress, and antioxidants: chronic diseases and aging." Arch Toxicol 97(10): 2499-2574.
Kanofsky, J. R. (1993). Singlet Oxygen in Biological Systems: A Comparison of Biochemical and Photochemical Mechanisms for Singlet Oxygen Generation. Oxygen Free Radicals in Tissue Damage. M. Tarr, Samson, F. (eds) Birkhäuser, Boston, MA.
Krasnovskaya, O., A. Naumov, D. Guk, P. Gorelkin, A. Erofeev, E. Beloglazkina and A. Majouga (2020). "Copper Coordination Compounds as Biologically Active Agents." International Journal of Molecular Sciences 21(11): 3965.
Molinaro, C., A. Martoriati, L. Pelinski and K. Cailliau (2020). "Copper Complexes as Anticancer Agents Targeting Topoisomerases I and II." Cancers 12(10): 2863.
Mucha, P., P. Hikisz, K. Gwozdzinski, U. Krajewska, A. Leniart and E. Budzisz (2019). "Cytotoxic effect, generation of reactive oxygen/nitrogen species and electrochemical properties of Cu(ii) complexes in comparison to half-sandwich complexes of Ru(ii) with aminochromone derivatives." RSC Adv 9(55): 31943-31952.
Rani, J. J. and S. Roy (2023). "Recent Development of Copper (II) Complexes of Polypyridyl Ligands in Chemotherapy and Photodynamic Therapy." ChemMedChem 18(8): e202200652.
Svanberg, K., N. Bendsoe, J. Axelsson, S. Andersson-Engels and S. Svanberg (2010). "Photodynamic therapy: superficial and interstitial illumination." J Biomed Opt 15(4): 041502.
Szewczyk, G., K. Mokrzynski and T. Sarna (2024). "Generation of singlet oxygen inside living cells: correlation between phosphorescence decay lifetime, localization and outcome of photodynamic action." Photochem Photobiol Sci 23(9): 1673-1685.
Tan, Y., S. Sun, D. Chen, H. Qiu, J. Zeng, Y. Wang, H. Zhao and Y. Gu (2021). "Light delivery device modelling for homogenous irradiation distribution in photodynamic therapy of non-spherical hollow organs." Photodiagnosis Photodyn Ther 34: 102320.

Round 4
Reviewer 3 Report
Comments and Suggestions for Authors
Since a systemic infection is being dealt with, it is not clear how PDT could be applied. A second issue is the well-known ability of Cu, to quench excited states necessiting the use of very high light fluxes to get a significant response. Why Cu would be chosen is never explained. Use of blue light also has a poor penetrating power and is readily scattered. The need for such a high light dose at the soret band is an indication of the inefficiency of this approach. For most PDT in vitro studies, longer wavelengths are used and at 300-500 mJ/sq cm light doses. So clearly this is a very inefficient process, likely resulting from the presence of Cu.
Hydrogen peroxide is a relatively stable molecule, so it is likely adequate to measure H2O2 formation after irradiation. It appears that singlet oxygen formation was measured during irradiation. How microbial infections would be treated is never adequately explained. What would be the target? It is not surprising that a relatively inefficient procedure might not have a significant effect on mammalian cells. PDT is usually directed at malignant cell types which tend to have upregulated LDL receptors. This form of therapy would not be successful if phototoxic effects on normal host cells was observed.
The major issues with this report are [1] no indication how a systemic infection might be treated, [2] the presence of Cu requires very high light doses in vitro, [3] it is not clear from section 4.3 whether SOSG was present during or after irradiation.
Author Response
We respectfully consider that all scientific concerns have been thoroughly addressed, and at this stage, we will refer the matter to the Academic Editor for final evaluation.